



# Assessing the impact of wind profiles at offshore wind farm sites for field data-enabled design

Rebeca Marini[1], Konstantinos Vratsinis[1], Kayacan Kestel[1], Jonathan Sterckx[1], Jens Matthys[1], Pieter-Jan Daems[1], Timothy Verstraeten[1,2], and Jan Helsen[1,3]

[1]Acoustics & Vibration Research Group / OWI-Lab, Vrije Universiteit Brussel, Pleinlaan 2, Brussel
[2]Artificial Intelligence Lab Brussels, Vrije Universiteit Brussel, Pleinlaan 9, Brussel
[3]Flanders Make @ VUB, Pleinlaan 2, Brussel

**Correspondence:** Rebeca Marini (rebeca.marini@vub.be)

**Abstract.** As wind turbines grow and wind farms become denser, more insight into real metocean conditions is essential for operational efficiency and load assessment. Light Detection And Ranging LiDAR) technology, which can substitute the use of meteorological masts, has garnered significant attention in the literature. However, it indirectly measures wind parameters, relying on assumptions and built-in algorithms. Wind field reconstruction (WFR) methods offer users greater control over
LiDAR measurements, enabling customised flow assumptions and parameter estimation. These measurements were taken during a measurement campaign on a wind turbine in the Belgian offshore zone. The WFR method has detected weather events, such as high shear, during the measurement campaign. These events are also linked to on-site weather conditions by using open-source metocean data. The findings align with the current literature on the correlation between events and weather conditions and the clear difference between wind profiling and a power law wind profile for loads design as proposed by
the International Electrotechnical Commission (IEC) standard. The results emphasise the importance of real measurements in understanding wind field characteristics, offering improved accuracy compared to standard assumptions, such as the IEC power law profile used for load design. This work underscores the value of real-life wind profiling for designing and operating wind farms in offshore environments.

## 1 Introduction

Through initiatives such as the European Green Deal and the EU Renewable Energy Directive, the EU strives to increase the share of renewable energy in its energy mix. With objectives such as carbon neutrality by 2050 and significant emissions reductions by 2030, the EU is at the forefront of accelerating the transition to renewable energy in its member states. These goals contribute to global climate action and position the EU as a leader in the transition to a low-carbon economy, fostering innovation and resilience amid climate challenges (Ritchie et al., 2023; EWEA Business Intelligence, 2015).

Although offshore wind farms generally incur higher construction costs than their onshore counterparts due to more complex cabling, transportation, and maintenance, they are gaining popularity due to their higher wind potential and environmental advantages (Bórawski et al., 2020). Technological advances in turbines have led to larger installations with higher energy yields, underscoring the need for optimisation of operational efficiency. An aspect of this optimisation is the better understanding



and estimation of the incoming wind field (Azad et al., 2011), which can impact the overall performance, power output and longevity of wind turbines. The latter is directly related to the aeroelastic loading that the wind turbines endure during their lifetime, which is connected to the wind profile approaching the wind turbine.

In wind energy, accurately measuring the wind speed and direction at the site is essential to optimise lifetime and energy output (Simley et al., 2011; Schlipf et al., 2011). Traditionally, monitoring and analysing wind parameters have relied on meteorological masts, but it can be costly and logistically challenging in an offshore site to install and maintain it (Hasager et al., 2008). In this context, Light Detection And Ranging (LiDAR), a remote sensing technology first used in the 1960s in other sectors, offers an alternative by eliminating the need for traditional masts (Mehendale and Neoge, 2020; Sharma et al., 2021). LiDAR is sampling the wind field, by using laser beams to detect aerosol movement, enabling the retrieval of wind velocity projections based on the Doppler effect (Trabucchi, 2020). This allows wind characteristics to be time-averaged at various spatial locations upstream and along the rotor area, improving wind farm operation.

While it is possible to conduct controlled experimental campaigns — such as those using dynamometers (e.g., at NREL, top left in Figure 1 (National Renewable Energy Laboratory, n.d.)) — or rely on simulations with validated models (Gebel et al., 2024) and even prototype field tests, it remains a need to further validate and explore the real, site-specific wind profiles that a wind farm will encounter. Moreover, drastic changes in environmental conditions, such as storms or extreme wind direction changes, can have a significant impact on turbine loading. To this end, these phenomena are typically taken into account during ultimate strength analyses (Commission and other, 2019). These real-life wind profiles, derived from field data, provide insights that cannot be completely replicated through controlled experiments or simulations alone. Such data reflects the actual atmospheric conditions, including localised effects of atmospheric stability, turbulence, and wind direction variability, which are essential for accurately assessing turbine performance and loading. By adding field data, it becomes possible to refine the wind field assumptions, ensuring that wind turbines are further optimised to handle both theoretical (i.e. more usual) and real-life conditions at a specific site, as suggested in Figure 1. This approach not only enhances the reliability of turbine designs but also contributes to improved energy yield predictions and the longevity of wind farm assets.

In this study, a methodology for resource assessment is investigated, and the factors for turbine load design are analysed. Multiple data sources, comprised of metocean data, LiDAR measurements and Supervisory Control and Data Acquisition (SCADA) data, are used and combined over long time horizons to analyse and validate key insights on the long-term steady-state environmental conditions at the site. To this end, wind-wave distributions, atmospheric stability, turbulence intensity and veer and shear profiles are extracted from the available data sources. Moreover, dynamic events, such as the veer and shear profiles during extreme events (e.g., storms), are automatically detected in the data on a large time horizon. This allows for a complete view of the resource aspects that are important for both fatigue and ultimate strength loading.

The paper starts by discussing the methodology in Sect. 2. Section 3 describes the different datasets used in the experiments. Section 4 investigates the experimental results, aiming to provide insights on the necessity of realistic wind profiles for employment in aeroelastic loadings assessment using simulations. Finally, Sect. 5 provides an overview of the current research.





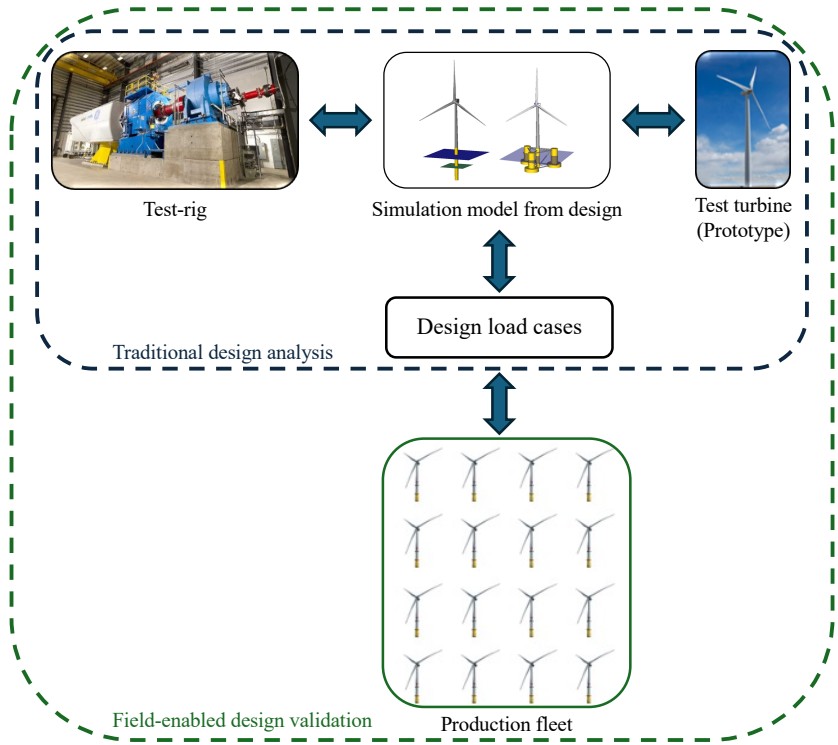

**Figure 1.** Diagram showing the interdisciplinary fields for designing load cases, not taking into account the possibility of assessing with the in-situ wind profiling.

## 2 Methodology

Research into the application of LiDAR technology in wind energy has increasingly focused on improving wind profiling (Mikkelsen, 2014; Hasager et al., 2013) and reducing the levelized cost of energy (LCOE) (Canet et al., 2020; Simley et al., 2020). This has been achieved through approaches such as increasing annual energy production, extending turbine lifetimes (Scholbrock et al., 2016), and reducing maintenance costs (Russell et al., 2023). While much of the existing research has centred on the use of vertical LiDAR systems for long-term measurement campaigns, the current study shifts the focus to the use of nacelle-mounted (horizontal) LiDAR systems. These systems offer the advantage of providing direct wind measurements along the rotor plane of the wind turbine, enabling new opportunities for performance optimisation.

Although LiDAR technology offers a viable alternative to conventional wind measurement methods, its use for accurately assessing wind parameters is contingent upon data processing techniques. These methods, known as wind field reconstruction (WFR), refine flow assumptions and enable the estimation of key wind characteristics. In this work, an established WFR method, the two-beam method (Wagner et al., 2014), is implemented to assess the wind field, laying the foundation for analysing critical atmospheric phenomena.



Wind profiling allows for the calculation of shear and veer; while the former is the change of wind speed with height, the latter is the change of wind direction. Optimising offshore wind farm operations requires understanding how weather events such as high shear and veer impact the wind turbine. For example, wind shear directly influences the energy extracted by the turbine rotor (Gao et al., 2021; Wagner et al., 2009) and can trigger shear-induced instabilities that lead to vibrations or damaging loads on the turbine structure (Gualtieri, 2016; Dimitrov et al., 2015). While wind veer generally has less influence

than shear on power production (Murphy et al., 2019) and aeroelastic loading (Robertson et al., 2019), its effects, particularly on larger turbines with increasingly significant directional differences across the rotor, remain an area of ongoing investigation (Tumenbayar and Ko, 2023).

Accurately characterising wind profiles is critical for reliable turbine operation, yet the current industry-standard power law approximation is more accurate only under specific atmospheric conditions (Bratton and Womeldorf, 2011). Deviations from

these conditions can lead to errors in energy production estimates (Fırtın et al., 2011) and turbine load predictions (Shaler et al., 2023). Additionally, anomalies in wind shear exponent often occur during extreme events such as thunderstorms and low-level jets, highlighting the limitations of static shear coefficients in complex meteorological conditions (Ryu et al., 2022).

This chapter outlines the methodology employed in the present work, which addresses these challenges by implementing LiDAR-based measurement and WFR techniques. For each method employed, the approach is explained and supported by

research in different literature.

## 2.1    Nacelle-mounted LiDAR

Wind field reconstruction (WFR) methods are techniques designed to estimate the wind field over a specific area, typically using data from one or more sensors. These methods are essential because LiDAR technology, which detects backscattered light along the line-of-sight of the emitted laser beam, has inherent limitations in capturing the full wind vector. This limitation,

often referred to as the "Cyclops Dilemma," creates ambiguity in single-line-of-sight measurements, making WFR challenging when relying solely on a monostatic LiDAR system (Guillemin et al., 2016). To overcome this, WFR can be achieved either by using multiple sensors targeting the same point in space or by positioning a single sensor measuring at different areas. At the same time, the former option is cost-prohibitive, and the latter relies on assumptions that simplify the wind field.

A WFR method requires at least two linearly independent line-of-sight measurements to accurately deduce the wind vector

components. The primary goal is to determine the horizontal wind speed and direction at multiple heights across the rotor area using LiDAR data. This study focuses on implementing the 2-beam reconstruction method, which operates under specific assumptions about wind flow characteristics, including horizontal homogeneity, either a two- or three-dimensional wind vector, and a predetermined shear profile (Borraccino et al., 2016). Previous implementations of the 2-beam reconstruction method, such as in (Schlipf et al., 2012; Wagner et al., 2014), have applied similar principles with slight methodological variations.

Figure 2a presents a top-view schematic of a wind turbine with a nacelle-mounted LiDAR (indicated by an orange dot), illustrating the setup for capturing line-of-sight measurements.





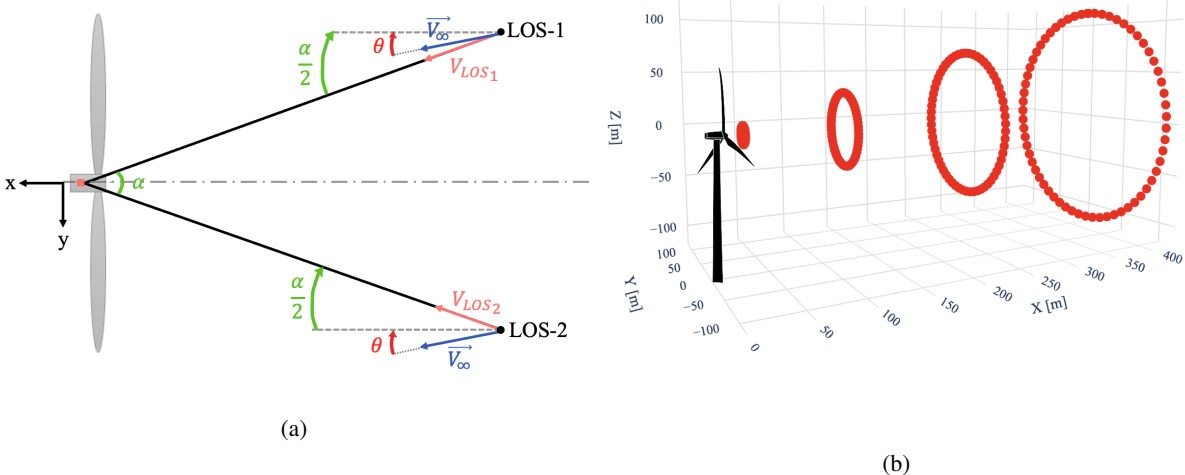

(a)

(b)

**Figure 2.** Nacelle LiDAR schematics for 2-beam reconstruction method implementation, in 2a, with a representation of four planes at different distances possible to measure the wind field, in 2b.

LiDARs can be separated into two main categories that depend on the laser source: continuous wave or pulsed laser beam. In the former configuration, the LiDAR will focus the beam and sample at a single distance from the wind turbine, with sampling frequencies ranging between 50 and 400 Hz. Meanwhile, the latter allows for a simultaneous multiple-distance backscatter

analysis with a light frequency emission between 10kHz and 20kHz. The presented research has employed a continuous-wave nacelle-mounted LiDAR from ZX LiDARs giving samples around the rotor at a 50Hz frequency.

The LiDAR used for the present work will measure circular profiles upstream of the wind turbine. At each line-of-sight measurement, the azimuthal position is registered concerning the centre of the probed circle. The measurements are averaged in 10-minute intervals and in space, creating a 64-averaged azimuthal position LiDAR, as seen in Figure 2b. Under specific

situations, such as when the laser beam hits a blade, data quality is compromised. Quality control is performed on the data to filter out poor-quality measurements where necessary. Furthermore, the chosen wind turbine is located in the first row of the wind farm, i.e., in the first row of the wind farm, such that for a large set of wind directions, the incoming wind field is undisturbed. This region coincides with the dominant wind direction at the wind farm's location, coming from the southwest.

## 2.2    Wind vector reconstruction

Through the schematic (Figure 2a), it is possible to visualise how the line-of-sight measurements (pink arrows $V_{los_1}$ and $V_{los_2}$) relate with the incoming wind field (blue arrows $\boldsymbol{V_\infty}$). Each line-of-sight vector will make an angle equal to the LiDAR's half-opening angle (green angles $\frac{\alpha}{2}$) with the x-direction. The LiDAR's manufacturer defines this angle as equal to 30 degrees ($\alpha = 30°$). Under the assumption that the incoming flow is homogeneous, as shown in the schematic, it will yield a flow misalignment (red angle $\theta$).





The 2-beam reconstruction method addresses the Cyclops dilemma by inferring the two-dimensional wind vector, $V_x$ and $V_y$, from two line-of-sight measurements. Using the calculated wind vector components, the wind speed magnitude ($V_\infty$) and wind direction misalignment ($\theta$) can be derived with, respectively, Eq. (1) and Eq. (2).

$$V_\infty = \sqrt{V_x^2 + V_y^2} \tag{1}$$

$$\theta = \mathrm{arctan2}(V_y, V_x) \tag{2}$$

For the current implementation, two key assumptions are made. First, the wind vector is considered two-dimensional, as the laser beam is assumed to be nearly horizontal, such that the vertical component (in the z-axis direction) becomes negligible. Second, horizontal homogeneity is assumed, meaning that the wind speed components remain consistent across different measurement locations, with no significant changes in flow characteristics.

The yaw angle of the wind turbine, $\psi$, must also be accounted for in calculating the wind direction. However, this adjustment

is usually minimal due to the wind turbine's automatic yaw adjustment according to the wind direction during normal operation. The 2-beam reconstruction method is a valuable tool for using LiDAR data to determine wind components and is especially useful for power performance analysis (Schlipf et al., 2012). Though relatively simple, this method yields critical insights into the wind conditions at the LiDAR location.

As the chosen LiDAR instrument measures along a circle in the rotor area, Eqs. (1) and (2) are applied at different heights.

Different volumes ($i$) will be created for the chosen heights, with a certain number of the measurements inside, as illustrated in Figure 3, in which the wind field is reconstructed with a matrix solution.

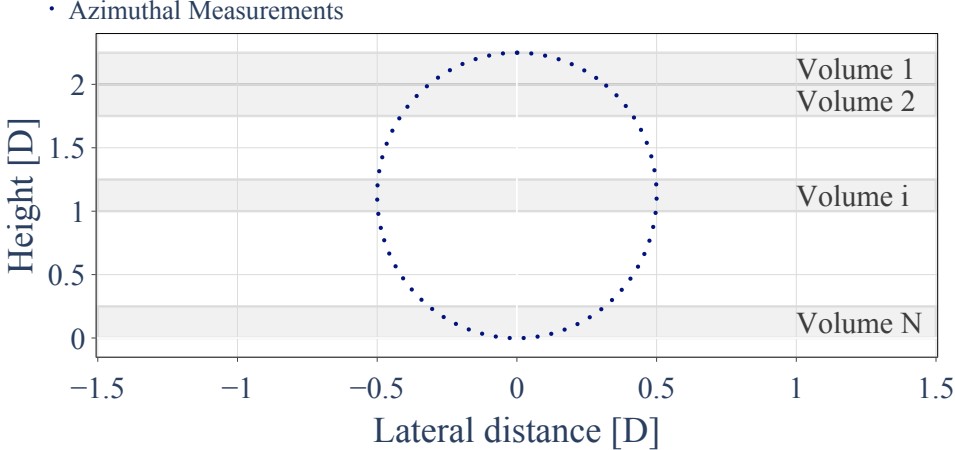

**Figure 3.** Reconstruction at different heights.





These events are thoroughly described in the literature as the correlation between events (Emeis, 2014), their seasonal and/or diurnal distributions (Yan et al., 2022), and their effect on power production (Gao et al., 2021; Murphy et al., 2019).

### 2.2.1 Shear

The shear is wind speed change along the height, and it can be normally defined by the power law, as in Eq. (3):

$$V_z = V_r \left( \frac{z}{z_r} \right)^\alpha \tag{3}$$

where $V$ is the wind speed at a specific height, $z$, and at reference height, $r$, and $\alpha$ is the shear exponent. Although power law is used to extrapolate wind speed in many cases, it has been proved that it cannot be as correct since shear is affected by numerous factors, such as turbulence intensity, atmospheric stability, and surface roughness (Yan et al., 2022).

High wind shear conditions can be associated with low-level jet events, where the wind velocity profile changes drastically below and above the jet height (Gadde and Stevens, 2021). For the current paper, detecting such events is based on fitting the estimated profile with the 1/7th standard one and not by definition of the event itself. The definition of such events can limit the profiles that the current research aims to analyse, i.e. more profiles can be detected with a correlation analysis with the standard profile than by the definition of specific events.

### 150 2.2.2 Veer

The veer is wind direction change along the height. In meteorology, it occurs due to the Ekmal spiral balance with Coriolis force, pressure gradient force and friction, advection due to thermal winds, and inertial oscillation (Gao et al., 2021; Brugger et al., 2019). Understanding wind veer is essential for several engineering applications, including but not limited to the evaluation of wind loads and as well the analysis of the wind turbine performance and wind power output (Shu et al., 2020).

Depending on the veer value ($\beta_{total}$), the wind flow is considered to be veering (positive veer) or backing (negative veer). While the former, which implies a clockwise rotation with height, is associated with positive gains in the power production estimation, the latter, with a counterclockwise movement, leads to power deficits (Murphy et al., 2019; Tumenbayar and Ko, 2023). In this paper, the veer is calculated according to Eq. (4) (Murphy et al., 2019):

$$\beta_{total} = \sum_{i=0}^{n} \frac{\left| \theta_{z(i+1)} - \theta_{z(i)} \right|}{z_{top} - z_{bottom}} \tag{4}$$

with $\theta$ the estimated wind direction and $z$ the correspondent height. Depending on its value, the wind flow is considered to be veering (positive veer) or backing (negative veer). While the former, which implies a clockwise rotation with height, is associated with positive gains in the power production estimation, the latter, with a counterclockwise movement, leads to power deficits (Murphy et al., 2019; Tumenbayar and Ko, 2023)



## 2.3 Weather conditions

The occurrence of high shear and high veer events will be associated with environmental conditions, such as air and sea temperatures, wind and wave directions, turbulence intensity and atmospheric stability. These are calculated using the different datasets retrieved for this study, such as the buoy and measurement pile data from a location near the wind farm. The wind-wave misalignment (Eq. 5) and air-sea temperature difference (Eq. 6):

$$\triangle\theta_{WW} = |\theta_{air} - \theta_{sea}| \tag{5}$$

$$\triangle T_{air-sea} = T_{air} - T_{sea} \tag{6}$$

where the subscripts $air$ and $sea$ describe the direction, $\theta$, and temperature, $T$, respectively, of the wind and the sea surface. These two conditions are then compared with the distribution of high shear and high veer events, creating a bridge between events and weather conditions. Furthermore, turbulence intensity is also calculated on a 10-minute interval (Eq. 7) with the 1-second SCADA data:

$$TI = \frac{\sigma_V}{\overline{V}} \tag{7}$$

where $\sigma_u$ is the standard deviation and $\overline{V}$ the mean of the wind speed.

### 2.3.1 Atmospheric Stability

In the lower layers of the Earth's surface (below 1 km), the interaction with the Earth's surfaces and buoyance forces affect the wind. The resistance of the atmosphere to these vertical movements is called atmospheric stability. This stability is heavily
influenced by thermal gradients and frictional drag, which can be induced by surfaces or wind shear (Wharton and Lundquist, 2012). The high or low atmospheric mixing causes the wind profile to resemble less with the power law profile, being only in near neutral conditions that this profile usually occurs (Sakagami et al., 2015).

This parameter can be defined according to different methodologies, either with direct or indirect measurements. The current paper computes the Obukhov Lenght (Eq. 9) with the use of the Bulk Richardson Number (Eq. 8) and classifies the atmosphere
into five classes: very unstable, unstable, neutral, stable and very unstable. (Holtslag et al., 2014). With the *Meetnet Vlaasme Banken* data, it is possible to calculate the Bulk Richardson Number, as in (Albornoz et al., 2022), with:

$$R_B = \frac{gz(T_a - T_s)}{V^2} \tag{8}$$

being a simplification for open sea conditions, $g$ is the gravitational acceleration, $z$ is the height above sea level, $T_a$ and $T_s$, respectively, the air and temperatures and $V$ the wind speed. After calculating the Bulk Richardson Number, it is possible to
estimate the Obukhov length, $L$, according to (Holtslag et al., 2014), with:





$$z/L = \begin{cases} \frac{10R_B}{1-5R_B} & if \quad R_B \geq 0 \\ 10R_B & if \quad R_B < 0 \end{cases} \tag{9}$$

The atmospheric stability is compared with the estimated shear and veer since these all influence how the wind profile reaches the wind turbine.

### 2.4 Non power-law events detection

The standard power-law profiles fail to capture some weather events, such as high wind speeds and rapid wind direction change (Debnath et al., 2021). However, these events are detectable with the wind vector obtained by applying the wind field reconstruction method in the rotor area. The detection is made using the mean absolute error, i.e., $MAE$, defined as:

$$MAE = \frac{\sum_h^{N_{\text{heights}}} |V_h^{\text{iec}} - V_h^{\text{est}}|}{N_{\text{heights}}}, \tag{10}$$

where $N_{\text{heights}}$ is the number of heights used across the rotor area, and $V_h^{\text{iec}}$ and $V_h^{\text{est}}$ are the wind speeds at height $h$ according
to the power law and estimated profiles, respectively. The power law is defined according to the International Electrotechnical Commission (IEC) standard 61400-3-1 (Commission et al., 2019).

In Figure 4, some divergent profiles are shown compared to the standard power law. These have $MAE$ values higher than 2. In the current research, these non-correlated profiles are associated with weather events, as these diverge from the power law profile with a low shear exponent that represents the average offshore wind profile.

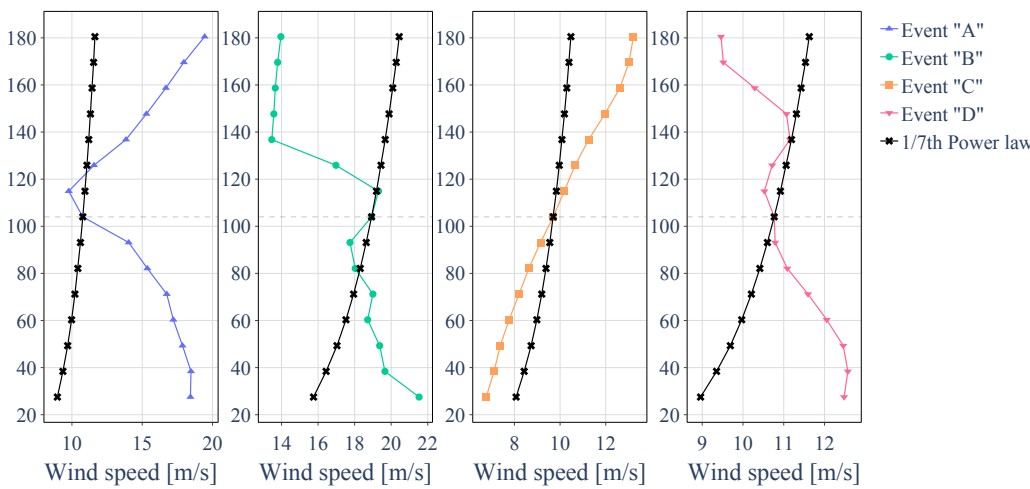

**Figure 4.** Estimated wind speed profiles for selected events deviating from the standard power law, compared to the standard profile (shown in black).



### 2.4.1 Extreme direction change

The IEC 61400-1 (Commission and other, 2019) standard defines the extreme direction change (EDC) events by defining the wind direction change magnitude, $\theta_e$, as:

$$\theta_e = \pm 4 \arctan\left( \frac{\sigma_1}{V_{\text{hub}}\left(1 + 0,1\left(\frac{D}{\Lambda_1}\right)\right)} \right) \tag{11}$$

The turbulence standard deviation, $\sigma_1$, is given by $I_{ref}(0.75V_{\text{hub}+b})$, with $I_{ref}$ being the expected value of turbulence at 15 $m/s$ equal to 0.16, for the wind turbine class in the study, and $b = 5.6\,m/s$. The turbulence scale parameter, $\Lambda_1$, is measured in meters as a function of the hub height of the wind turbine, being equal to $42m$ for the current study. Finally, $D$ is the rotor diameter. The magnitude of the wind direction change is dependent on the wind speed at hub height, $V_{\text{hub}}$, so for each wind speed, the magnitude will change. Furthermore, the ramp-up of an EDC event should follow $\pm 0.5\theta_e(1 - cos(\pi t/T))$, with the duration of the event, $T$, equal to 6 seconds.

### 3 Dataset

The current research used three different data sources; their overview is presented in this section. Firstly, it uses nacelle-based LiDAR data obtained from a measurement campaign in a wind farm in the Belgian offshore zone. In addition, the SCADA of the same wind turbine has been used to estimate the turbulence intensity. Finally, the environmental conditions are obtained using an open-source dataset from the Belgian government, the *Meetnet Vlaamse Banken* data.

### 3.1 Nacelle-mounted LiDAR

A LiDAR has been deployed from February of 2023 until mid-March of 2024, with interruptions due to maintenance between June and August of 2023. Thus, the analysis is performed on 12 non-consecutive months in the aforementioned period. The sensor was installed on a wind turbine in the Belgian offshore zone, located in the first row of a wind farm, as seen in Figure 5a. In Figure 5b, a photo of the LiDAR measuring the data used in this research is shown.





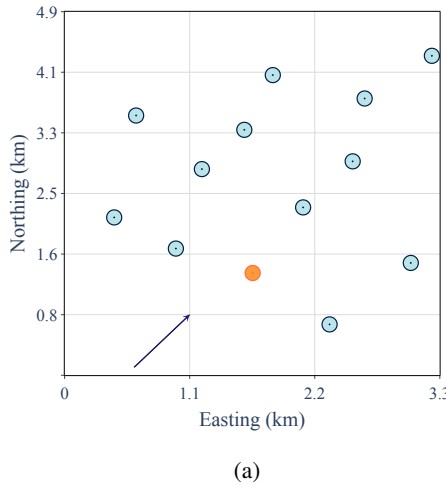

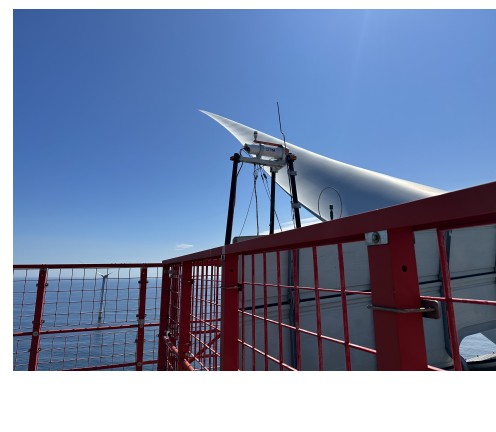

|(a)|(b)|

**Figure 5.** The schematic of the wind turbine used during the measuring campaign 5a, with the photography of the installed LiDAR in 5b.

The deployed LiDAR provides a model based fit-derived (FD) wind components for the 10-minute averaged dataset at the different distances and heights for hub height. These include the horizontal wind speed and the vertical wind shear exponent, derived from proprietary software from LiDAR's manufacturer. These FD wind components are used to compare and validate the experimental setup, i.e., the LiDAR's installation in the nacelle. After deploying the measurement campaign, the LiDAR's FD records were compared with the wind data captured by the sensors installed by the wind turbine manufacturer, the SCADA

system.

### 3.1.1   Data filtering

All reconstruction results are obtained at a distance of 2.56 times the rotor diameter upstream of the turbine. Except for the seasonality analysis, the data is filtered to consider only the freestream wind direction, spanning from the south to the west. This directional filtering is essential for isolating relevant wind conditions, as non-freestream measurements would negatively

influence the accuracy of the reconstruction process. As described in (Marini et al., 2024), the reconstructed wind vector from the non-freestream directions has a greater error when compared with freestream ones.

The raw LiDAR data is filtered using the validity criteria specified by the manufacturer, retaining only valid line-of-sight measurements. After filtering, approximately 70% of the data is considered valid for further analysis. Similarly, SCADA data is filtered according to the yaw misalignment, and if the wind turbine is curtailed, the dataset is reduced to 60% of its original

size. The data is then averaged into 10-minute intervals and binned across 64 azimuthal points, as detailed in Section 3. Finally, an additional filter is applied to isolate data where the wind originates from the freestream and local dominant wind direction, between south and west. These approach the wind turbine for roughly 52% of the time during the measurement campaign. Since the turbine is located in the front row, it experiences freestream conditions, meaning turbine wake effects are absent





in this directional sector. However, the absence of wake data presents a limitation for wind field reconstruction in the wake
direction.

## 3.2    Environmental data

This study utilises data from the *Meetnet Vlaamse Banken* dataset provided by the Flemish government in Belgium (De Wolf,
1986). The dataset offers meteorological and oceanographic data, measured in buoys or measuring piles on the Belgian North
Sea. For the current work, several measurements were retrieved from both instruments, including wind speed and direction
measurements, wave height, and sea and air temperatures.

   The data is retrieved from the Westhinder buoy and measurement pile around 30 km from the coast (en Kust, 2024). Although
it does not coincide with the exact LiDAR's position, it is assumed that these will be correlated with the wind farm area. While
the dataset provides high-quality data, time resolution is a significant limitation. Since the results of the reconstruction are
10-minute averaged, and this data has a 30-minute resolution, the gaps were addressed using a linear interpolation method.

From this dataset, it is possible to calculate the wind-wave misalignment, air-sea temperature difference and atmospheric
stability. These results are compared with the distribution of events, making a relation between the wind profile and the in-situ
weather conditions.

## 4    Results

The methodology described in Section 2 is applied to the LiDAR, SCADA, and Metocean data sets presented in Section 3. In
this section, the results obtained in the analysis are shown and briefly discussed. First, the results from the measurement cam-
paign are filtered and shown in terms of wind resource and seasonal effects. This is followed by validating the reconstruction
method with a time series compared with SCADA and the manufacturer's reconstruction. The atmospheric stability calculated
with *Meetnet Vlaamse Banken* data is shown and briefly analysed. Results for shear and veer are shown afterwards. Finally, the
non-power-law events are shown, and initial conclusions are drawn.

## 4.1    Wind resource

In Figure 6, the wind speed and wind direction distributions, estimated with the wind field reconstruction method, are shown. A
wind resource assessment is the first step for wind farm planning and optimisation, as the wind speed and direction distribution
describe the wind energy resource and the estimation of power generation, which is also used for the economic viability study of
construction, size and layout of the wind farm. However, it is also necessary to perform continuous wind resource assessment
to track its temporal variation and evolution accurately. The data available in the current research allows us to assess the
wind resources exactly where several wind farms have been constructed. Furthermore, this analysis enables validation of the
reconstruction method, as the parameters are identical with historical data (Quante and Colijn, 2016), the local measurements
with SCADA and reanalysis data as ERA5. For the duration of the measurement campaign, the estimated average wind speed
is about 12 m/s, and the dominant wind direction is southwest (225°).





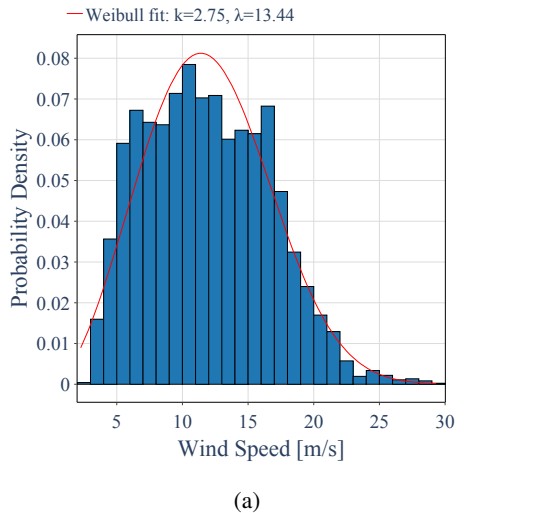
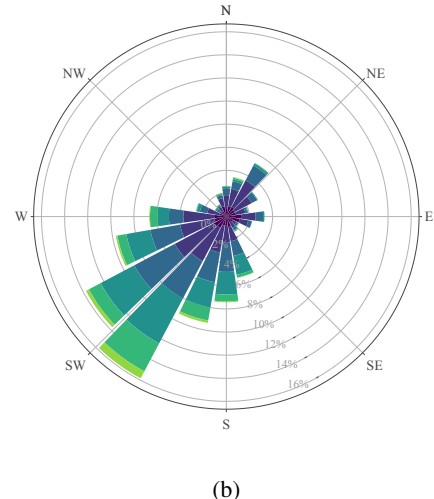

(a)           (b)

**Figure 6.** Histogram with weibull distribution of wind speed 6a and wind direction wind rose 6b.

### 4.1.1 Seasonal analysis

The measurement campaign has been deployed for more than one year, which allows for an investigation of seasonal trends. In Figure 7, it can be observed that winter (Figure 7a), autumn (Figure 7d) and summer (Figure 7c) have relatively similar behaviour with regards to incoming wind conditions. However, a prevalent NE wind direction is present during spring (Figure 7b), which is not the case for the other seasons; nevertheless, the prevalent wind speeds from the NE direction are significantly lower in speed than the dominant wind direction (southwest).





Wind Speed Bins [m/s]:
- [2-5]
- [5-9]
- [9-13]
- [13-17]
- [17-21]
- [21-25]

(a) winter

(b) spring

(c) summer

(d) autumn

**Figure 7.** Wind roses of estimated and measured wind field parameters during the whole period of the measurement campaigns in which the LiDAR was mounted on the first row (a) and the second row (b).

For the seasonal analysis, the data used is not filtered in the freestream wind direction, which can reduce reliability in the remaining directions (Marini et al., 2024). This is expected since, in those cardinal directions, there might be a high probability that the flow is not homogenous due to wake, as the wind is disturbed by the presence of upstream wind turbines, as seen in Figure 5a. The presence of wakes is associated with disturbances in the measurements, especially so far away from the turbine where the measurement circle can be large enough for one line of sight to be in wake and the other not. In the disturbed wind field, the assumptions of the 2-beam method do not hold as strongly. The seasonal analysis indicates variations in wind patterns that can impact turbine performance, highlighting the importance of continuous monitoring and adaptation to changing conditions.






## 4.2 Wind field reconstruction

In Figure 8, the reconstruction at several heights, from closest at 27.5 m above sea level (a.s.l.) in the lightest red to furthest at 180.5 m a.s.l. in the darkest red, is compared with the estimated hub-height wind speed from ZX LiDAR (blue) and the SCADA measurements (black). With these reconstructions, estimating the weather events of interest, such as veer and shear, is possible.

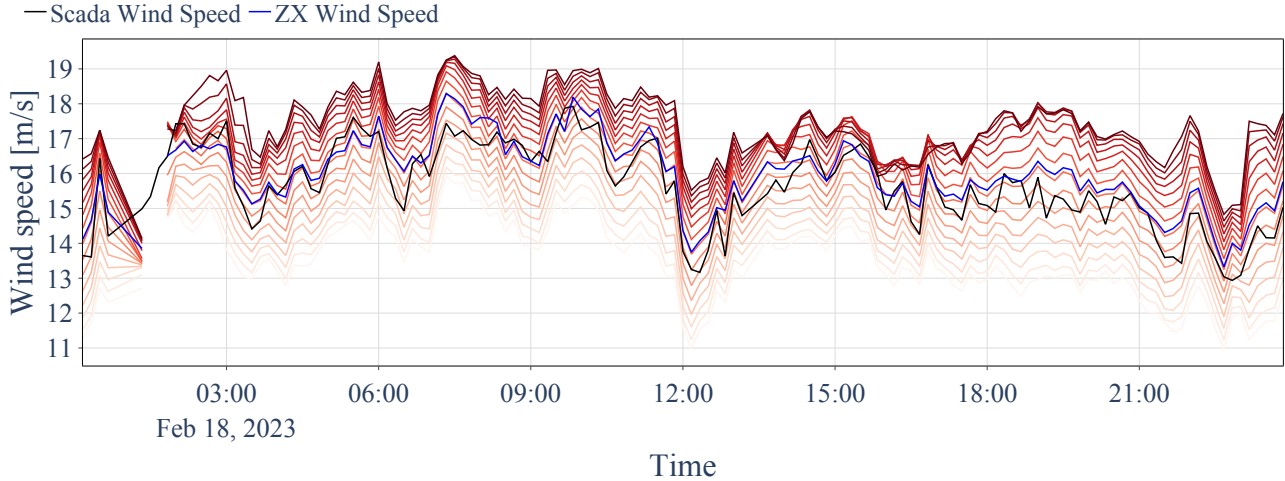

**Figure 8.** Reconstruction at different heights compared with the ZX "black-box" reconstruction (blue) and the SCADA measurements (black).

A validation process, which enhances confidence in the measurements, has shown that the 2-Beam and fit-derived recon-
struction have a high correlation with SCADA, with an $R^2$ of 0.98 and $R^2$ of 0.94, respectively. These strong correlations show the method's effectiveness in providing precise wind speed estimations. However, the fit-derived method relies on proprietary software from the manufacturer, limiting user control.

The 2-beam reconstruction methodology offers enhanced flexibility regarding the granularity and the specific heights at which the reconstruction is performed, as it permits the selection of reconstruction heights, making it applicable across the en-
tire dataset. Additionally, while the proprietary reconstruction is constrained to 10-minute intervals, the implemented method-ology supports reconstruction intervals as small as 1 second.

## 4.3 Atmospheric Stability

The atmospheric stability distribution throughout the measurement campaign is shown in Figure 9. The difference between the 10-meter air temperature and the sea surface temperature was calculated using data retrieved from Meetnet Vlaamse Banken to
estimate atmospheric stability. This dataset was filtered to match the period when LiDAR measurements were available. Atmo-spheric stability classes were defined using intervals of the Monin-Obukhov length (Equation 9), which provides a framework





for categorising stability into very stable, stable, stable-neutral, neutral, unstable-neutral, unstable, and very unstable conditions (Sathe et al., 2010). The Monin-Obukhov length is a widely used parameter for classifying atmospheric stability derived from heat, momentum, and atmospheric stratification fluxes.

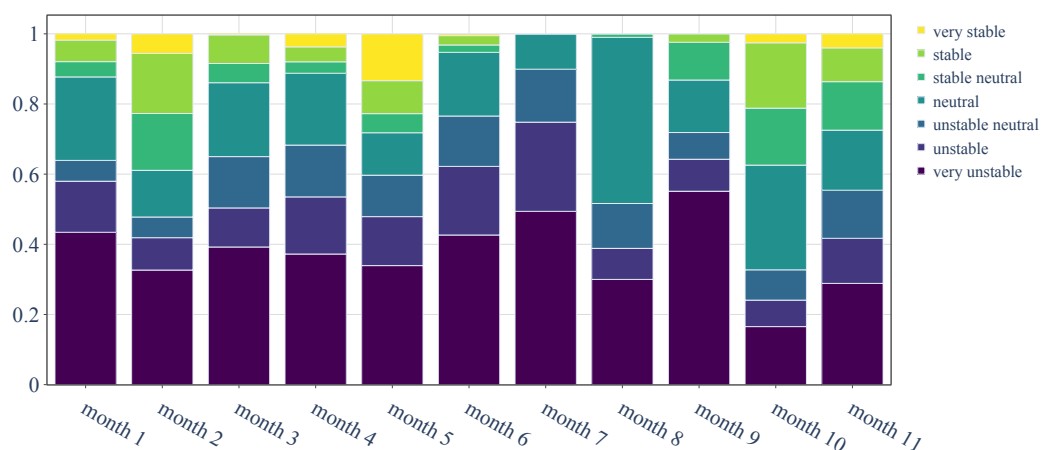

**Figure 9.** Atmospheric stability distribution along the measurement campaign.

The results in the figure indicate that neutral conditions dominate the atmospheric stability distribution over the course of the campaign. This finding is consistent with expectations for an offshore environment, where the relatively uniform thermal properties of the sea surface lead to stable vertical temperature gradients and low variability in atmospheric stratification. Neutral or slightly unstable conditions are typical in offshore locations due to the thermal inertia of the ocean, which limits extreme surface temperature deviations compared to the air above. This contrasts with land-based environments, where diurnal

and seasonal heating and cooling patterns can result in frequent extreme stable or unstable stratification.

Seasonal trends are also apparent in the figure, with stable and very stable conditions slightly more common in colder months, particularly during the winter. This pattern aligns with scenarios where colder air moves over warmer sea surfaces, leading to a stable stratification. Conversely, during warmer months, slightly unstable or unstable conditions become more frequent due to warmer sea surface temperatures heating the air above, promoting vertical mixing. However, extreme unstable

conditions remain rare, reflecting the relatively moderate heat fluxes over the ocean compared to land-based environments. These results align well with findings in previous studies on offshore atmospheric stability. For example, (Sathe et al., 2010) has reported similar distributions, highlighting the dominance of neutral conditions and the reduced frequency of both highly stable and highly unstable conditions offshore.





## 4.4 Shear

The average wind shear coefficient, calculated using the formula seen in Eq. 3, for the duration of the measurement campaign, was $0.062$. The wind profile is usually extrapolated using the power law with a constant shear exponent of $0.14$, as defined by the IEC guidelines (Commission et al., 2019). Using this extrapolation with a constant value is not ideal since the wind profile can be affected by several factors, such as the local roughness and atmospheric stability (Yan et al., 2022). Although the conditions usually fit the standard power law in an offshore context, that is not always the case. The average wind shear

coefficient, calculated using the formula seen in Eq. 3, for the duration of the measurement campaign, was $0.062$.

Around 82% of the positive estimated wind shear will fall under the $1/7$ value, meaning if the power law profile is assumed, the IEC standard will be more conservative in the majority of cases. Since the study takes place in an offshore location, where the atmospheric stability remains close to neutral (see Figure 9) and with higher wind speeds, a power law profile is expected (Wharton and Lundquist, 2012). Moreover, around 7% of the duration of the measurement campaign, the wind shear resulted

in a negative value, which implies a non-power law profile the larger absolute value it has. As defined by the IEC standard 61400-1 (Commission and other, 2019), the extreme wind shear exponent equals $0.2$, and for the current research, the values above it constitute a high shear event.

In Figure 10, the red dashed line highlights the $1/7$ value within the shear coefficient distribution. With the LiDAR measurements it is possible to reconstruct the wind speed at several heights, so it is also possible to calculate the value of the power

exponent using Eq. 3, in which the assumption of the power law profile is taken.

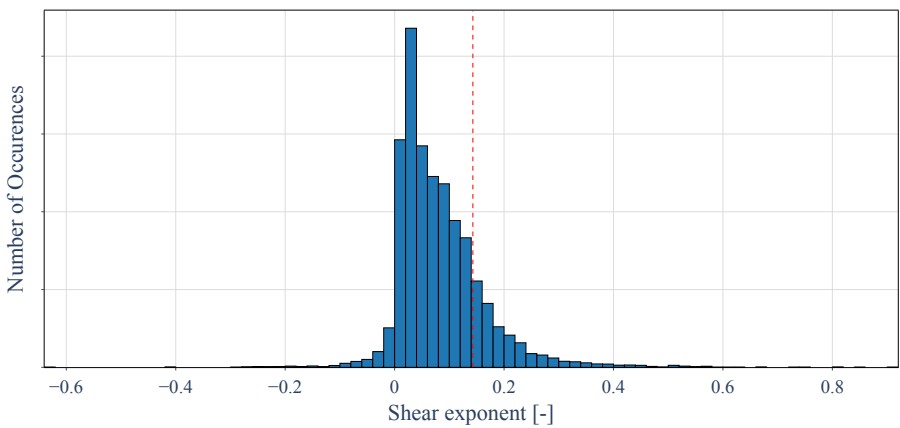

**Figure 10.** Power law exponent (shear $\alpha$) distribution. The red dashed line denotes the $1/7$ value.

In Figure 11, the monthly occurrences where the absolute value of the shear exponent is higher than $0.2$ is shown. Furthermore, these are displayed for different wind speed categories: above-rated power (>14 m/s) and between rated and cut-in wind speeds.



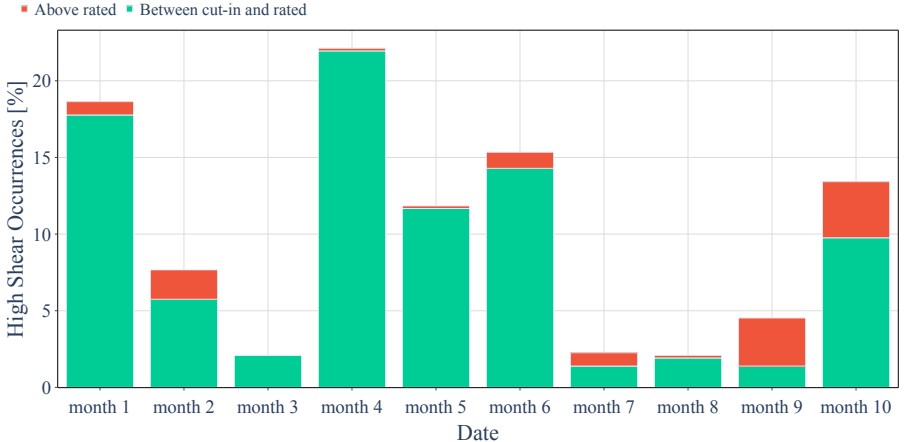

**Figure 11.** High shear occurrences throughout the measurement campaign.

### 4.4.1 Relation with environmental parameters

Atmospheric stability is not the only phenomenon that influences wind shear, as discussed in section 2. As seen in Figure 12, other conditions will also affect the wind shear approaching the wind turbine. While wind-wave misalignment, or the difference between wind and wave directions, in Figure 12a, appears to hold less influence than the other analysed weather conditions. When looking at the air-sea temperature difference, in Figure 12b, the distribution of high shear events shows a deviation for when the air is warmer than the sea surface. As for turbulence intensity, calculated with SCADA data, the high shear events

also deviate slightly from the original distribution, causing more events to occur in lower values.

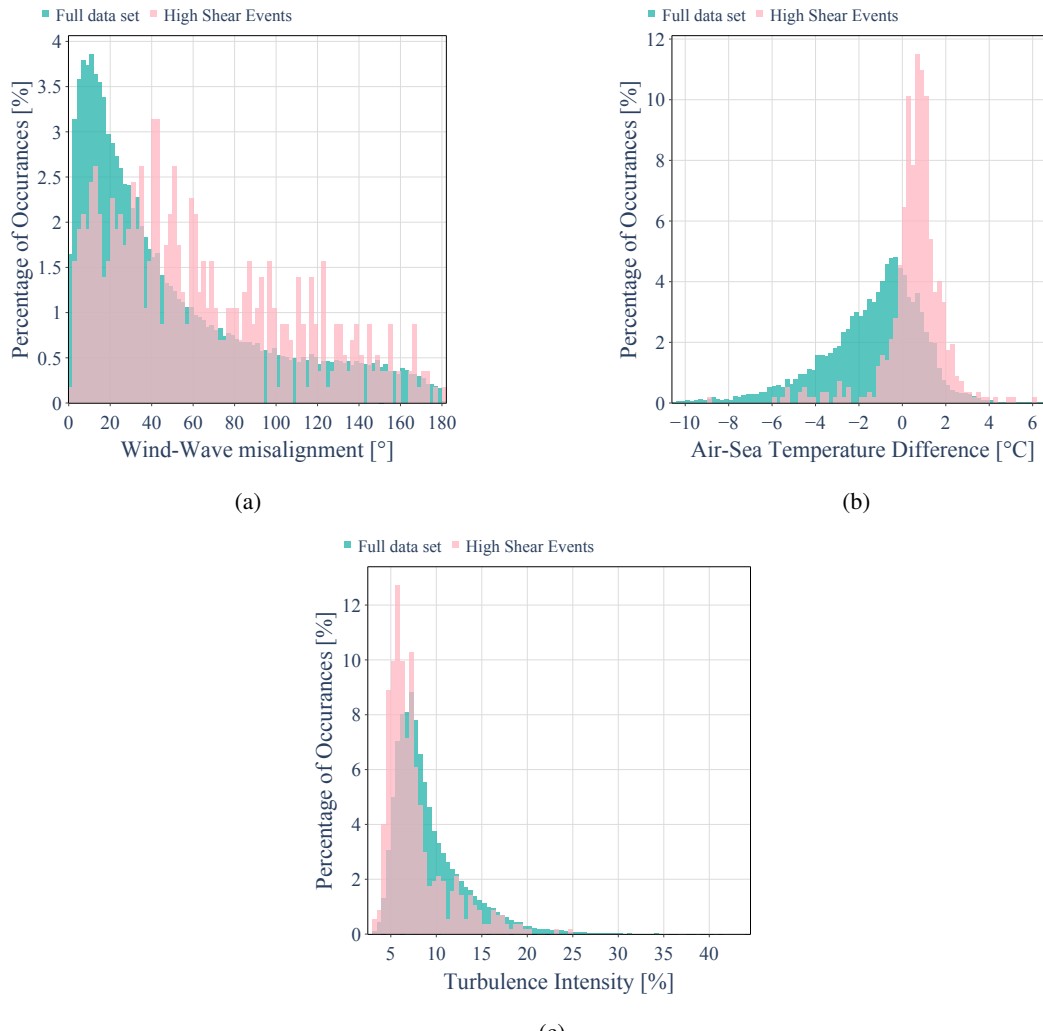

**Figure 12.** High shear events compared with the distribution of the met-ocean data, as wind-wave misalignment (12a), air-sea temperature difference (12b) and turbulence intensity (12c).

The analysis of wind shear reveals distinct patterns under varying atmospheric stability conditions, as in Figure 13. High positive wind shear primarily occurs during stable conditions associated with warmer air temperatures. This, in turn, correlates with broader patterns observed in the relation between air-sea temperature and high shear distributions (Figure 12b). On the other hand, high negative wind shear is predominantly found under unstable conditions. However, the two highest bins of negative shear occur mostly between stable and neutral conditions.





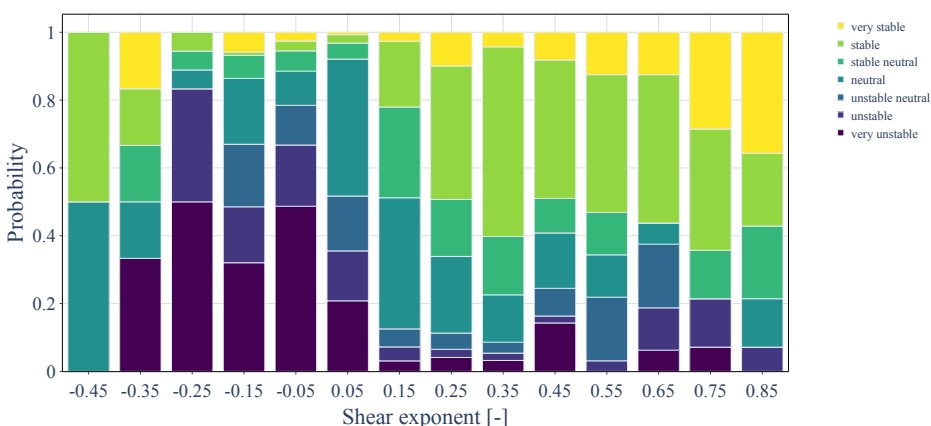

**Figure 13.** Shear exponent distribution along the atmospheric stability classes.

Despite the influence of atmospheric stability on wind shear, the IEC standard (Commission and other, 2019) prescribes the use of shear models such as the power law wind profile for load calculations, which are independent of stability. However, it is well established that atmospheric stability also affects wind shear, as seen in the current study. Recent studies have explored the impact of stability on wind resource assessments, turbine performance, and fatigue load estimations, emphasising the need
to incorporate stability effects in such analyses (Holtslag et al., 2014).

## 4.5 Veer

Veer is the change rate of wind direction along the height and is calculated according to Eq. 4, which can derive both negative and positive values. In Figure 14, the veer distribution is shown along the zero-line to display the cases of backing and veering better. For the duration of the measurement campaign, it was estimated around 70% of the time to be veering and 30% to be
backing, which is expected for a northern hemisphere offshore location (Gao et al., 2021).



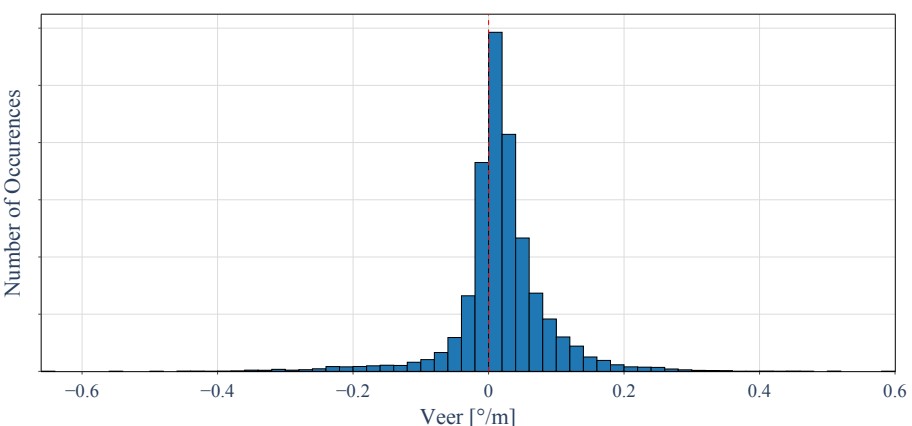

**Figure 14.** Veer ($\beta_{bulk}$) distribution. The red dashed line denotes the 0 value, separating veering and backing.

In Figure 15, the monthly occurrences of high veer, i.e., absolute value above 0.2, and separated by the categories mentioned above, are shown. All months of the measurement campaign have high veer events, contrary to high shear. Furthermore, the contrast between "Month 1" and "Month 10", which are the same months but in different years, is visible. With this, it is possible to see that the high veer also correlates with weather conditions since, although in the same season, in 2024, there was

an increase in the overall temperature and recorded rainfall (Walker, 2024).

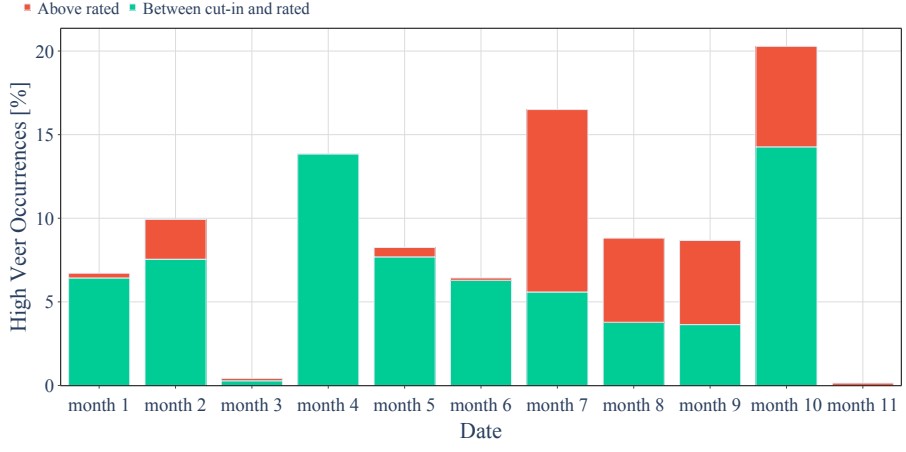

**Figure 15.** High veer occurrences throughout the measurement campaign.




### 4.5.1 Relation with environmental parameters

Figure 16 shows the relationship between weather parameters and the occurrences of high veer. The air-sea temperature differ-
ence (Figure 16b) appears to have a more significant impact on this event than wind-wave direction misalignment (Figure 16a).
It appears that turbulence intensity also plays a role (Figure 16c), as higher turbulence levels can degrade measurement accu-
racy and, by extension, impact the reliability of the wind field reconstruction. Therefore, it remains a variable when interpreting
the data, though its influence may be less direct than temperature effects. Understanding these events in relation to turbulence,
temperature gradients, and wind-wave interactions provides deeper insight into the mechanisms driving high wind veer events.

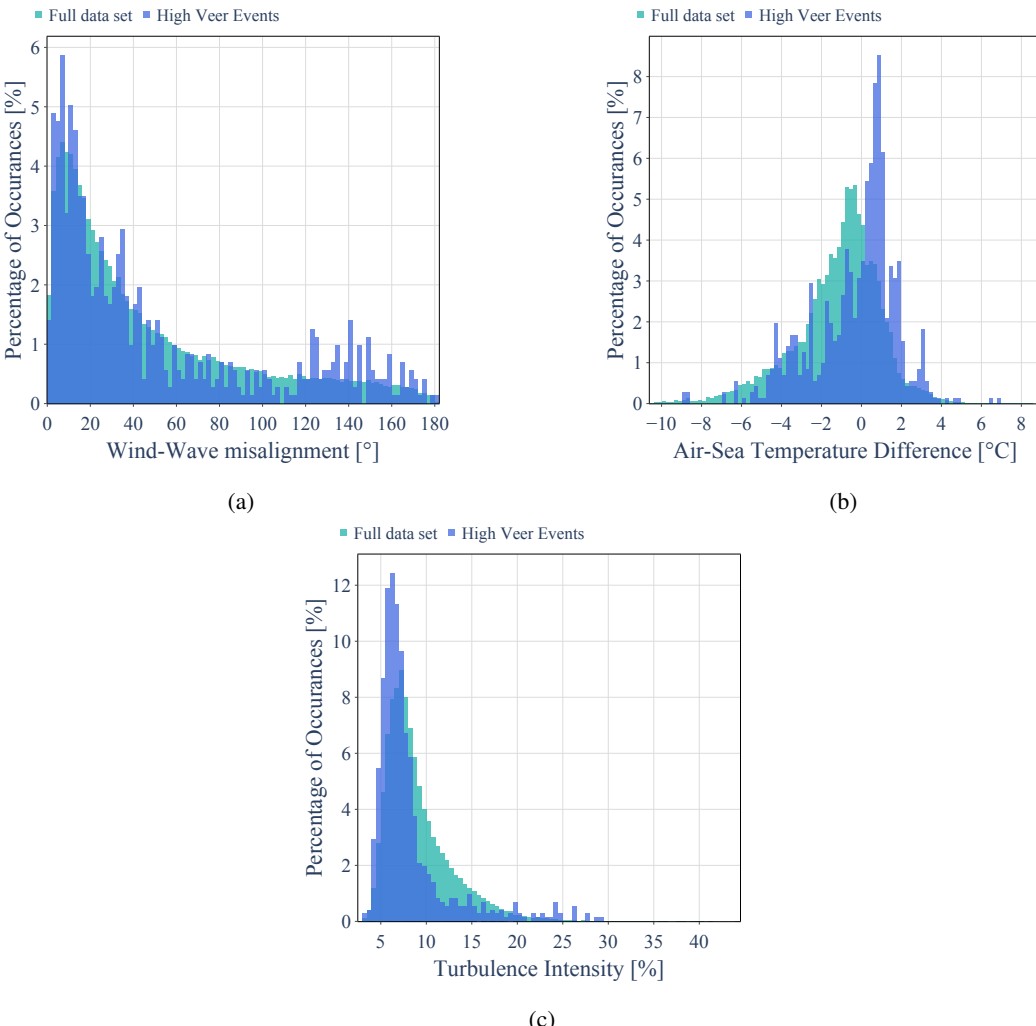

**Figure 16.** High veer events compared with the distribution of the metocean data, as wind-wave misalignment (16a), air-sea temperature
difference (16b) and turbulence intensity (16c).





Figure 17 shows the distribution of wind veer across different classes of atmospheric stability. Wind veer, characterised by a clockwise rotation of wind direction with height, tends to occur more frequently under stable atmospheric conditions. In contrast, wind backing (counterclockwise rotation) is typically associated with unstable conditions (Tumenbayar and Ko, 2023). The occurrence of wind veer is driven by a complex interplay of forces, including the pressure gradient, Coriolis force, and friction due to surface roughness. Under idealised conditions, Ekman's theory predicts a wind veer of approximately 45° in the atmospheric boundary layer. However, this can vary between 15° and 40° in practice, depending on atmospheric stability Brown et al. (2005). Furthermore, it has been studied that wind veer is reduced in cases of cold advection or thermal instability, highlighting the significant role that temperature gradients and stability play in modulating wind veer behaviour (Shu et al., 2020).

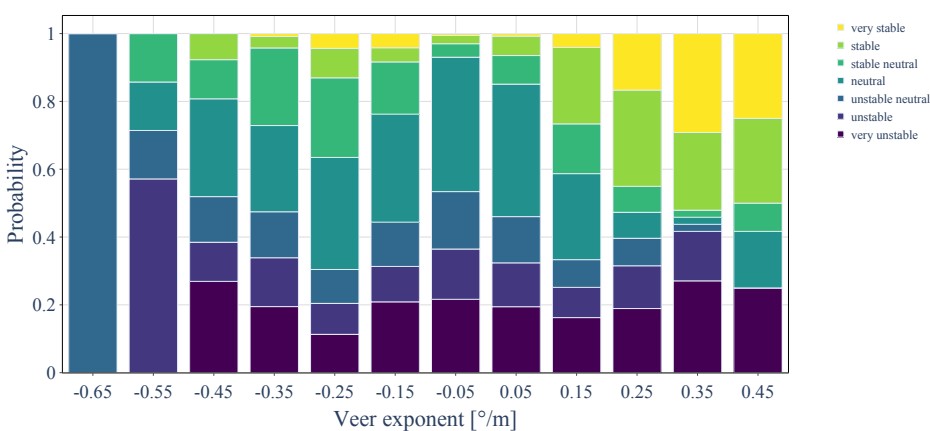

**Figure 17.** Veer distribution along the atmospheric stability classes.

In (Brugger et al., 2019), it has been shown that both in simulations and field measurements, the wake shape is skewed in the direction of the wind veer. The presence of veer, or backing, will influence both the power prediction (Murphy et al., 2019) and the loads on the wind turbine (Robertson et al., 2019). Furthermore, with the growth of wind turbines, the veer effect is less studied, so it is pertinent to consider veer in the multitude of studies related to offshore wind energy, either for designing, constructing or monitoring.

### 4.6 Non-power-law events

The non-power-law events are detected and further compared with the IEC standard. In addition, during the measurement campaign, it was possible to capture some registered storms in the Belgian offshore zone, for example, *Ciáran* during October/November of 2023. The reconstruction made during these periods is shown and analysed in comparison with the IEC





standard assumption of the power-law profile. Furthermore, the wind veer is also analysed and compared with the extreme direction change events detected during the same periods.

In Figure 18, an example is shown of how the power law profile assumption might not be ideal for some weather events, namely storms. The standard profile, reconstructed based on the hub height wind speed measurement, is compared with the
estimated wind profile, displaying a significant difference from reality. This estimated profile occurred during *Isaack* storm in February of 2023.

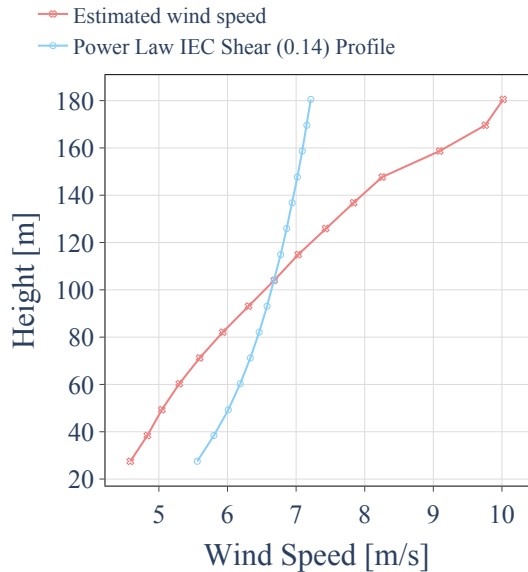

**Figure 18.** The estimated wind field (in red) compared with the IEC standard power law profile (in blue).

### 4.6.1 Storms

The storms discussed in this section are presented with time series snippets of wind speed at different heights, where lighter to darker colours represent increasing heights. When examining the estimated wind speed profile distribution over a full day,
as shown in this section, it is evident that the IEC standard assumption holds for most of the day, as seen in Figure 10. Additionally, wind speed profiles for the duration of each snippet are provided, with colours transitioning from the earliest to the latest timestamp.

During the storm *Ciarán* (in November of 2023), there were records of very high wind speeds, reaching around 100 km/h on the Belgian offshore coast; a snippet of the storm is shown in Figure 19. In the duration of the snippet, it is possible to see
that the power law profile occurs for the majority of the time. This further shows that high wind speed is not the only parameter influencing the wind profile.





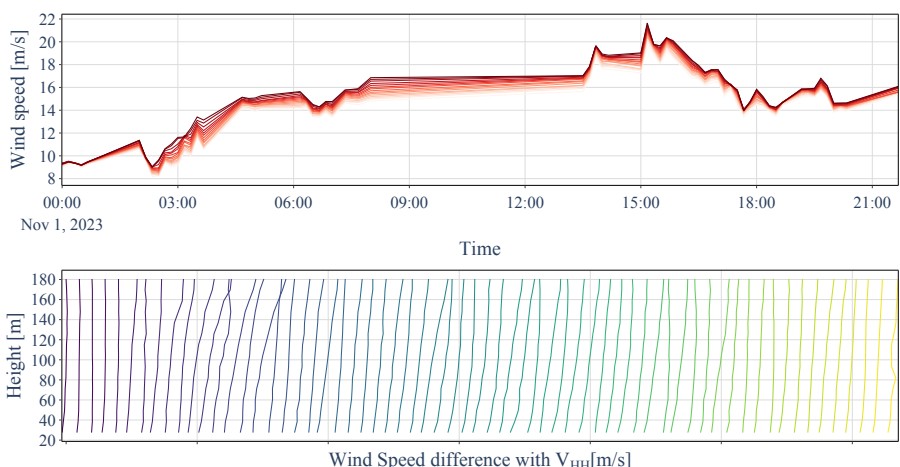

**Figure 19.** Time series of wind speed at different heights (top) with wind profiles (bottom) during *Ciarán* storm.

In Figure 20, a portion of one day during *Babet* (in October of 2023) storm is shown. While this storm passed by the Belgian coast, it had less impact than anticipated, affecting the UK and Germany. However, during this period, there was an increase in the average wind speed, and some of the estimated wind profiles appear unrelated to the power law profile.

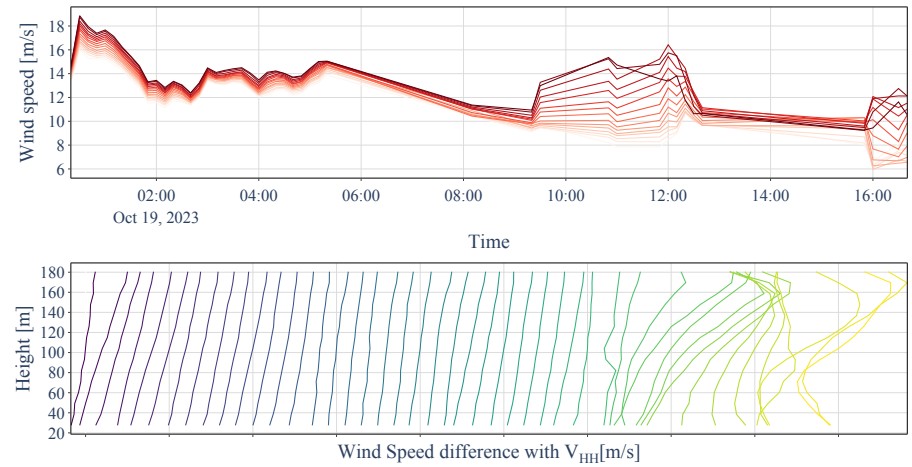

**Figure 20.** Time series of wind speed at different heights (top) with wind profiles (bottom) during *Babet* storm.

While in Figure 21, a portion on *Jocelyn* (in January of 2024) storm is shown. With higher wind speeds than the previous storm, this storm significantly impacted the wind profile. Moreover, the recorded wind speeds for this snippet are fairly close





to the ones shown for *Ciáran* storm. However, the profiles during *Jocelyn* have very little correlation with the standard power law.

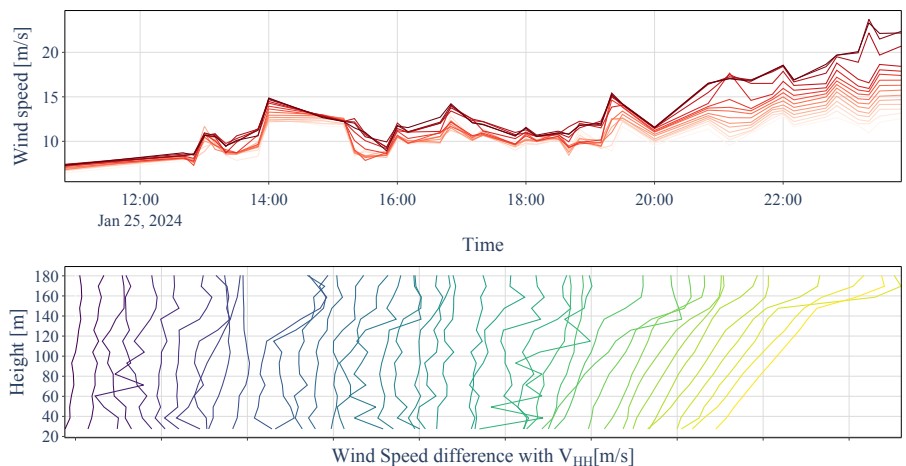

**Figure 21.** Time series of wind speed at different heights (top) with wind profiles (bottom) during *Jocelyn* storm.

Different recorded storms, such as *Babet*, *Ciarán* and *Jocelyn*, will have different impacts on the estimated wind profile. As
there is not one single factor that influences the wind profile, storms show that wind profiling estimation becomes imperative for more realistic input conditions for analysis such as such as a computational load assessment analysis.

**4.7   Extreme direction change**

The SCADA-based yaw misalignment analysis was conducted using data collected over the same period as the measurement campaign, with higher availability due to maintenance interruptions affecting the LiDAR. Access to SCADA data is particularly
valuable in capturing events characterised by extreme direction changes (EDCs), which are useful for understanding dynamic loading conditions on wind turbines. Although wind field reconstruction is unavailable during these short-duration EDCs, comparisons can be drawn between SCADA-detected EDC events and the daily average wind speed profiles to contextualise directional shifts within broader diurnal trends.

According to the IEC 61400-1 standard (Commission and other, 2019), an extreme direction change is defined as a wind
directional difference occurring over six seconds. The magnitude of this directional difference is calculated concerning the specific wind turbine under study, with equations provided in the standard. Detailed methodologies for calculating these changes are outlined in Section 2. This approach enables an analysis of yaw misalignment and contributes to understanding wind direction variability in harsh weather conditions. The following figures show the estimated wind veer in the blue circles and the EDCs identified by the SCADA data in the red-dashed lines. Furthermore, these figures correspond to the same days as those
analysed in the previous section.





Figure 22 shows that not so many EDCs were identified during *Ciaran* storm, although the wind veer changed from veering to backing a few times. The scattering is expected during such a storm, with metocean conditions changes, such as pressure, that affect the wind veer.

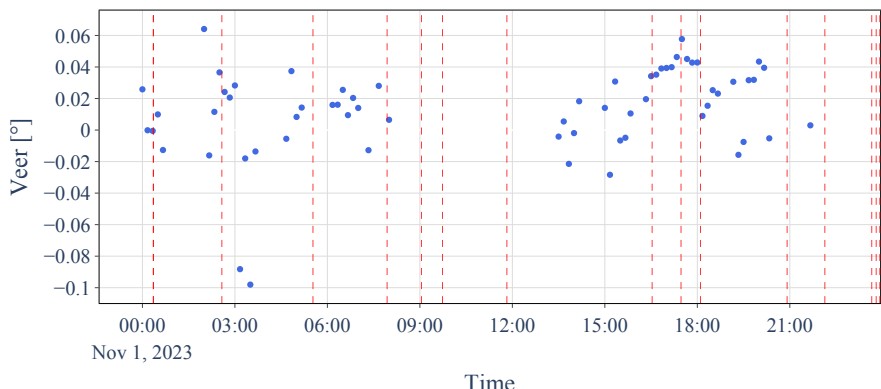

**Figure 22.** Time series of a day of the veer (blue circles) with the events of extreme direction change (red dashed lines) during *Ciarán* storm.

As for storm *Babet*, in Figure 23, the EDCs events start to occur around the same mark that the wind profile loses its correlation with the power law. This is accompanied by an increase, and then rapid decrease of wind veer, coinciding with the type of low-pressure areas cyclones have.

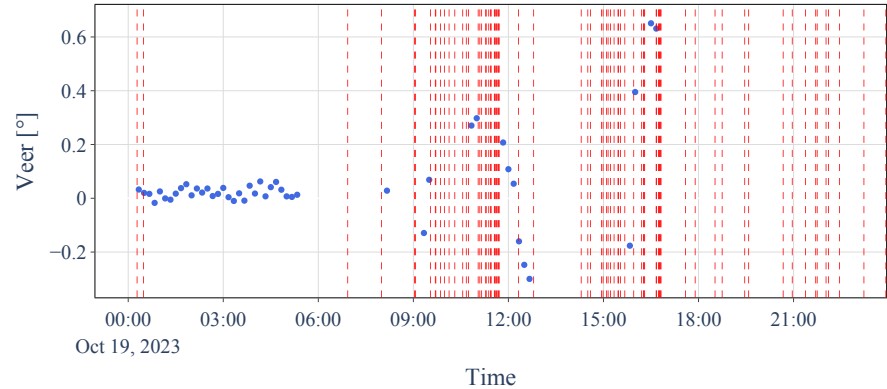

**Figure 23.** Time series of a day of the veer (blue circles) with the events of extreme direction change (red dashed lines) during *Babet* storm.





In Figure 24, *Jocelyn* storm's wind veer and EDCs are shown. While there is a change from backing to veering, there was almost no recorded EDC during this period. This suggests that although EDCs and veer might be associated, one does not necessarily imply the presence of the other.

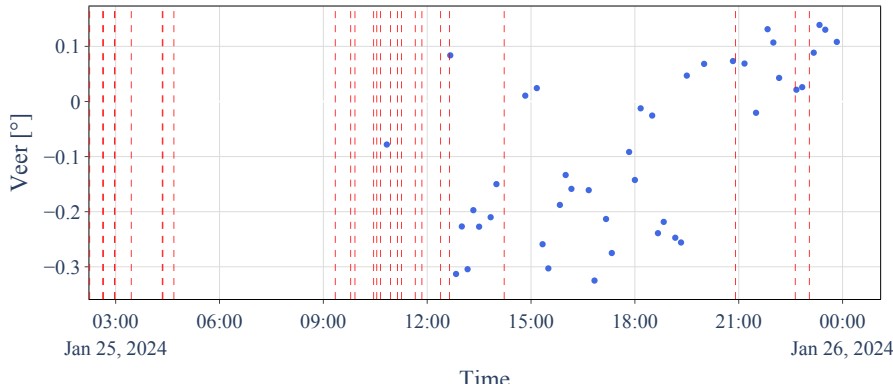

**Figure 24.** Time series of a day of the veer (blue circles) with the events of extreme direction change (red dashed lines) during *Jocelyn* storm.

The combination of reconstruction data, weather conditions, and SCADA has proven valuable for defining the incoming wind field and allows for creating more realistic wind field profiles for use in applications such as load design cases.

## 5   Conclusions

Following a measurement campaign on an offshore wind turbine, the measured data must be analysed and interpreted using appropriate methodologies. By incorporating additional data sources, such as measurement pile data and SCADA, a realistic

wind profile distribution for a specific location can be created. Knowing how to analyse and reproduce real-life wind profiles and their relation to environmental conditions allows, for example, creating a more realistic load case design.

    To estimate wind profiles, LiDAR measurements can be used as input in wind field reconstruction (WFR) methodologies to derive wind characteristics. The 2-beam WFR method uses two line-of-sight measurements to estimate the wind speed magnitude and direction misalignment. This approach is tested and validated through field experiments conducted within the

wind farm in the North Sea off the coast of Belgium.

    The methodology employed in the current research has allowed the exploration of wind characteristics along various heights, enabling the estimation of shear and veer in the wind field. These parameters, which cannot be estimated only with SCADA measurements, contribute to a more comprehensive understanding of the wind's behaviour. In addition, it was possible to analyse the relation between these parameters by using local weather measurements, with *Meetnet Vlaamse Banken* data. In

parallel, the comparison with atmospheric stability consolidates the correlation that shear and veer have with weather conditions. Finally, it was also possible to correlate the events with turbulence intensity by having access to SCADA.





Throughout the literature, it is known that wind shear and veer impact several areas related to wind turbines, from power production to the loads experienced by the wind turbine. These conditions can lead to increased structural stress, potentially shortening the lifespan of turbine components and affecting overall performance. Understanding the relationship between

atmospheric stability, shear, and veer is critical for optimising turbine design and operation in offshore environments and improving load prediction models.

The current measured-based wind profiling estimations offer a more accurate and realistic basis for designing load cases and evaluating power performance, complementing the assumptions outlined in the IEC 61400-1 standard. While the IEC standard provides a reliable framework under most operating conditions, improving wind profiling remains essential, particularly for

analysing extreme events that challenge conventional design limits. Enhanced profiling in these simulations will contribute to more robust load assessments and performance evaluations under various operating conditions.

Finally, despite the limitations of the 2-beam method, the robustness and reliability of the methodology are further demonstrated. When compared with SCADA data and *Meetnet Vlaamse Banken* measurements, the estimated profiles align well with expectations for the given location and meteorological conditions. However, further investigation is necessary for its

implementation under wake conditions and for real-time estimations, as it could, for example, be utilised for control purposes.

*Author contributions.* Rebeca has made the data curation and formal analysis, as well as the writing, revision and editing of the original draft and the final manuscript. Konstantinos, Kayacan, Jonathan, and Jens conceptualized and conducted the experimental work to generate part of the data used in this study. Pieter-Jan, Timothy, and Jan provided supervision, validated the results, and contributed to the review and editing of the manuscript. Finally, Jan has secured the funding necessary for this work.

*Competing interests.* The authors declare that they have no conflict of interest.

*Acknowledgements.* This research was supported by funding from VLAIO (Flemish Agency for Innovation and Entrepreneurship) through intercluster SIM-Blue Cluster ICON Rainbow and SBO CORE. The authors moreover, acknowledge the support via the Flemish Government under the "Onderzoeksprogramma Artifici¨ele Intelligentie (AI) Vlaanderen" program. This work used large language model softwares for spelling and grammar checks.





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
