# Peer review of "Assessing the impact of wind profiles at offshore wind farm sites for field data-enabled design"

_Wind Energy Science, 2025_

## Author Comment (AC1)

**Reply to anonymous referees**

We would like to thank all reviewers for their time for thoroughly analysing our manuscript and their constructive comments. We have carefully considered each comment and provide below our answers. Our manuscript has been revised accordingly and we believe that the revision will address all referees' comments.

In summary, we have rewritten parts of the abstract and introduction, rearranged the manuscript to improve reading logic (such as the description of the dataset is now presented before the methodology), and the results section was also extended with improvements, namely on the atmospheric stability discussion. Furthermore, the captions of Figures were modified, to improve their interpretability while reading the manuscript.

This document replies to every comment made by each of the anonymous referees, by order of appearance. Replies to anonymous referee #1 are in pages 2 to 8, to anonymous referee #2 in pages 9 to 15, and to anonymous referee #3 in pages 16 to 29.

**Anonymous Referee #1**

https://doi.org/10.5194/wes-2025-9-RC1

**Comment:** This article focuses on analyzing the wind conditions located in the Belgian Sea, by combining nacelle LiDAR data, wind turbine SCADA data, and site buoy marine meteorological data. On the level of research significance, I think this article is worth publishing. Because these wind conditions can be references for the design and control of offshore wind turbines, and reveal the limitations of the current turbine design standards. Overall, the authors have conducted a higher quality data analysis and data presentation. However, from the perspective of an academic paper, I think the writing logic of the article can be further improved. For example, certain content that should appear in the results section is presented in the methods section. For a data analysis type of article, I think an early introduction in the second chapter on how the data was collected, what data was used for the analysis, what key parameters the authors wanted to present, and so on, would have helped to improve the logic and readability of the article.

**Reply:** Thank you for the insightful summary of the paper and the general remarks on how to improve it. Following the suggested changes in this comment, we have now moved the "Dataset" section to appear before the "Methodology". As mentioned, this has improved the writing logic of the manuscript. Furthermore, we would like to thank for the detailed comments, as there are lapses that were not detected in the internal review process.

**Comment:** The probe volume effect has not been mentioned in this paper. For the continuous wave lidar analyzed in this work, the probe length can not be ignored when focusing at a far distance. The measured LOS speed is not point like but contaminated by LOS speeds over different heights and longitudinal distances. some of the distances in the prove volume near the rotor, it might be affected by the induction zone and making the homogeneous flow assumption used by the authors invalidated. This should be clarified or analyzed.

**Reply:** We have added context for the probe volume effect, as it is very relevant for the current work. However, we have decided to not extend the discussion further than a paragraph, as this topic does not reflect the work presented.

**Changes:** "However, lidar data are also subject to limitations, including the so-called probe volume effect, which can influence the accuracy of turbulence intensity estimation (Sathe et al., 2011b; Pauscher et al., 2016). That said, previous studies have demonstrated good agreement in mean wind speeds between nacelle-mounted lidars and reference masts in similar setups (Peña et al., 2017), suggesting that the influence of probe volume on mean flow characterisation is minimal in such contexts."

**Comment:** Abstract: I suggest pointing out what lidar measures (spatially averaged LOS speed) to give readers who do not know about lidar a basic idea of why lidar does not directly measure wind parameters.

**Reply:** We have changed the abstract during the review process, namely this sentence has been improved to convey a better basic idea of a lidar.

**Changes:** "It is a sensor that spatially averages the Doppler effect response from the aerosols in the atmosphere, and so, it relies on assumptions and built-in algorithms to provide wind parameters."

**Comment:** Abstract: The abstract should be reconstructed. It should be read as a mini paper indicating the motivation, the data collected, the method applied to analyze the data, and finally the conclusions (preferably with key indicators, e.g., how many samples the measured wind profile is significantly different from the log profile).

**Reply:** We have rewritten the abstract to be clearer and more concise.

Changes: New abstract added to the manuscript.

**Comment:** *Line 25, Can you add some references on how the wind profile can affect the turbine loads?*

**Reply:** We have added a reference "Robertson, A. N., Shaler, K., Sethuraman, L., and Jonkman, J.: Sensitivity analysis of the effect of wind characteristics and turbine properties on wind turbine loads, Wind Energy Science, 4, 479–513, 2019.", as it discusses the effect of wind characteristics on the wind turbine loads, and have concluded that "shear is [one of the] most sensitive parameters for turbine loads".

**Changes:** "Wind profiles, which are defined by parameters such as shear, veer, and turbulence intensity, directly impact energy production estimates (Wagner et al., 2009; Murphy et al., 2019), turbine loading (Gualtieri, 2016; Dimitrov et al., 2015; Robertson et al., 2019), and ultimately, structural longevity."

**Comment:** *Line 34, How the averaged wind characteristics can improve wind farm operation should be further explained.*

**Reply:** We have re-written the sentence to be clearer the statement on how the use of lidar can be an opportunity to obtain more accurate wind conditions at site and on hub-height, together with two references supporting it.

**Changes:** "These [lidars] systems provide line-of-sight (LOS) wind speed measurements across the rotor plane, offering new opportunities to more accurately characterise wind conditions at the turbine level (Simley et al., 2020; Russell et al., 2023)."

**Comment:** Line 35 to 45, There is an IEC-61400-13 standard which requires turbine manufactures to validate the turbine loads. This standard also recommends that nacelle lidar can be used for load validations of offshore turbines if the lidar system can provide a TI equivalent to anemometer measurement. The difference in TI by lidar and anemometer is still an ongoing research topic. Successful applying nacelle lidar for load validation has been demonstrated by https://doi.org/10.1002/we.2385. I suggest the authors to look into IEC-61400-13 and https://doi.org/10.1002/we.2385 as they are highly linked to the current study.

**Reply:** The suggested work has been a great addition to the literature review for this work. The paragraph has been altered to better reflect the author's intention at this moment of the manuscript,

which is the advantage of having field data, as it enables a validation of the load cases with a more accurate wind profile. Moreover, this profile does not have to come from lidar necessarily, but it is the field data we have at the moment.

**Changes:** Paragraph changed with the re-organisation of the introduction, so it does not have a direct change as it reads differently from the reviewed version.

**Comment:** The citation in Line 40 (Commission and other, 2019) is a bit strange. I recommend to check all citations and preferably add hyper links to them.

**Reply:** We introduced hyperlinks to the manuscript, and they work for us. In case there are still any issues, we will refer to the publisher to ensure they are enabled. In addition, we have checked all citations and have corrected where necessary. In the 'changes' to this reply, we have included the corrected citation mentioned in the comment.

**Changes:** "To this end, these phenomena are typically taken into account during ultimate strength analyses (IEC, 2005)."

**Comment:** *Line 47 a methodology for resource assessment -> for site condition assessment.*

**Reply:** We have made the correction as the scope of the manuscript is indeed site condition and not resource.

Changes: "In this study, a methodology for site condition assessment is investigated, (...)".

**Comment:** *Line 56. Normally the conclusions are given at the end. Maybe use ":an overview of points observed "*

**Reply:** We have removed the last sentence, as it became clear that it is not necessary, nor a usual practice.

Changes: Sentence removed.

**Comment:** *Line 63, direct wind measurements ->line-of-sight speed measurements.*

**Reply:** The correction has been written, which makes the sentence clearer and more precise.

**Changes:** "These [lidars] systems provide line-of-sight (LOS) wind speed measurements across the rotor plane, (...)".

**Comment:** Most of the sentences between Section 2 and Section 2.1 are background knowledge about lidar and wind field reconstruction and are not related to the method proposed by the authors. The authors should consider putting this content in the introduction or a subsection.

**Reply:** We have reviewed the writing logic of the manuscript, in which it was decided to move the information in the beginning of Section 3 (originally Section 2) to the "Introduction" chapter.

Changes: See reviewed manuscript.

**Comment: The caption of Figure 2 does not read clear.**

**Reply:** We have altered Figure 3 (originally Figure 2) to show the top view schematic, that shows the relation between the wind vector and line-of-sight measurements, and the 2D volumes schematic (originally in Figure 3).

**Changes:** "Figure 3. The left panel shows the top view schematic of the nacelle-mounted lidar (pink square) of the implemented 2-beam reconstruction method, with the wind vector represented in the blue arrow ( $V_{\infty}$ ), the projected wind vector in the LOS in the salmon arrows ( $V_{los1}$  and  $V_{los2}$ ), and the angles of the lidar's opening and misalignment represented, respectively, by the green  $\gamma$  and red  $\Delta$ . The right panel shows the schematic of the circle created with 64-averaged azimuthal positions projected in two dimensions, normalised with the wind turbine's rotor. Light grey illustrates how the circle is separated into volumes (i) used for the reconstruction at that respective height."

**Comment:** *Line 110, is it possible to provide a map of the measurement plane indicating the dominate wind direction and turbines? Also, the turbine rotor size should be mentioned.*

**Reply:** Due to the use of confidential data, the rotor size cannot be divulged as the wind farm needs to be anonymous. The right panel of Figure 2 (originally Figure 5) has displayed the information about the dominant wind direction and the relative location of the neighbouring wind turbines, so is now showing the relevant information earlier in the manuscript. The dominant wind direction has been made more explicit in the results chapter.

**Changes:** "For the duration of the measurement campaign, the hub-height estimated average wind speed is about  $12 \text{ ms}^{-1}$ , and the dominant wind direction is southwest ( $225^{\circ}$ ).".

**Comment:** *Line 125, the authors say that the beams are nearly horizontal. This contradicts with Volume N in Figure 3.*

**Reply:** Although the beams are nearly horizontal at a hub-height, it is indeed not true for the other heights. We have removed this statement as we focus on several heights for the reconstruction process within the manuscript.

**Changes: Sentence removed.**

**Comment:** Line 131-132, Firstly, the wind turbine yaw misalignment can be 10 degrees for typical *MW* level turbine. With the homogeneous flow assumption, one gets the wind direction relative the nacelle (lidar) orientation. To obtain the wind direction in the inertial coordinate system, one still needs the nacelle direction. Please clarify.

**Reply:** The oversight on the SCADA data used for the current work has been correct, with the addition of a subsection (2.2) dedicated to explaining the data and making it clearer that the wind turbine's position is used to estimate the wind direction.

Changes: "The yaw angle of the wind turbine, obtained with the turbine's SCADA, (...)"

**Comment:** *Line* 161: *theta* -> *theta\_z*.

**Reply:** We have corrected the theta to read the correct variable from equation 6 (originally equation 4).

**Changes:** "with  $\theta_z$  the estimated wind direction and (...)"

Comment: Line 160 to Line 164 repeats the contents in Line 156-159.

**Reply:** Instance repeated before equation has been removed, as the information is more relevant after presenting equation. Thank you for your attention while reviewing the manuscript.

**Changes:** Repeated sentence has been removed in the beginning of section 3.1.2 (originally section 2.1.2).

**Comment:** Equation 4, not all variables are explained.

**Reply:** All variables are defined in the paragraph after presenting equation 6 (originally equation 4).

**Changes:** "with  $\theta_z$  the estimated wind direction and z the correspondent height, n, with top and bottom being, respectively, the highest and lowest heights."

**Comment:** *Line 171: The space in the beginning should be avoided.*

**Reply:** There was an oversight with the space after the equations, all instances have been now correct.

Changes: Corrections made throughout the manuscript.

**Comment:** *Line 173 to Line 175 how the TI is being calculated? By the nacelle anemometer or LOS speeds, please clarify.*

**Reply:** The turbulence intensity is calculated using the SCADA from the turbine. The oversight has been corrected by adding a subsection (2.2) explaining the dataset and how it is used for the current work. Further clarity is added during the methodology, highlighting the use of SCADA data for the calculation of turbulence intensity.

**Changes:** "For the turbulence intensity, the 1-second SCADA data from the same wind turbine is used to perform the calculation in a 10-minute interval by applying (Eq. 7):"

Comment: Figure 4 should appear at the "Results" section.

**Reply:** Figure 4 shows examples of events with associated wind profiles, to demonstrate how the methodology can detect various cases of interest. However, we believe that the caption of the figure insufficiently captures this intent, and we decided to clarify this.

**Changes:** Four distinct profiles, in blue, green, yellow, and pink, that can be captured by the implemented methodology of non-power-law profile detection. These exemplify different deviations of captured profiles compared to the 1/7 power law, shown in black.

**Comment:** *"Meetnet Vlaamse Banken data" is explained in Section 3 but it firstly appear at Line 185.*

**Reply:** We have re-organised the manuscript, namely by having the data presented before the methodology, which solves this comment by first presenting the "Meetnet Vlaamse Banken data" and only after showing its usage.

Changes: Manuscript re-organised.

**Comment:** The caption of Figure 7 does not read clear, please provide more details.

**Reply:** We have corrected the caption of Figure 6 (originally Figure 7), as the original caption was erroneously present in the manuscript.

**Changes:** "Figure 7. Seasonal wind roses of the measured wind data for non-consecutive months between February 2023 and March 2024."

Comment: Line 232, can you explain why 2.56D is chosen? To avoid rotor induction effect?

**Reply:** As you mention, the 2.56D is chosen to avoid the rotor induction effect. It became evident that this information was not clear in the manuscript; it has been added.

**Changes:** "All reconstruction results are obtained at a distance of 2.56 times the rotor diameter upstream of the turbine, avoiding the rotor induction effect."

**Comment:** Figure 8: not all lines are provided with legends.**

**Reply:** We agree that the lines should be described in the figure. However, to prevent visual pollution, instead of adding the lines in the legend, we decided to clarify the depicted trends both in the caption of the figure and in the text.

**Changes:** "Figure 8. Reconstruction at different heights compared with the ZX "black-box" reconstruction (blue-cross symbol) and the SCADA measurements (black-circle symbol). The redcoloured lines indicate the reconstruction at different heights, with the lightest red indicating the lowest a.s.l. and the darkest red the highest a.s.l. heights."

**Comment:** *Line 316, please specify the figure number*

**Reply:** We have corrected the sentence in order to have the figure number that it refers.

Changes: "Seasonal trends are also apparent in Fig. 9 (...)"

**Comment:** Section 4.6, it would be great to show the non-power law MAE histogram as well

**Reply:** Figure has been added as it is a visual information of great relevance.

Changes: See Figure 18.

**Comment:** Figure 4, the most left chart looks like a wake profile. The authors should provide further information on these Events, e.g. wind direction, atmosphere stability, TI, wind wake misalignment on so on.

**Reply:** We agree that missing information is of relevance to explicitly show how the profiles and weather conditions are interconnected should be added.

**Changes:** "All events are within the south and west wind direction domain, as it is the region where the estimated wind profile comes from a freestream state. In terms of turbulence intensity, events "A", "B" and "C" have a relatively low intensity with, respectively, 5%, 7% and 6%, while event "D" has 24%, which is a very high turbulence occurrence. Finally, all events have different atmospheric stability classifications, with "A" being neutral, "B" very unstable, "C" very stable, and "D" an unstable event."

**Comment:** I suggest that the authors put the overview of the measurement campaign (basically Section 3) after introduction (Section 1) to improve the logic of this manuscript.

**Reply:** The sections have been changed, with the data being presented before the methodology, as it is a clearer reading/writing logic for the work presented in the manuscript.

Changes: See reviewed manuscript.

**Anonymous Referee #2**

https://doi.org/10.5194/wes-2025-9-RC2

**Comment:** I have strong difficulties understanding what the scientific objective of your study is, the research question(s), and what exactly you want to address with the analysis you present in your study. Right now, the manuscript reads as a pure description (a technical report) of a measurement campaign performed with a nacelle lidar on a turbine at the Belgian Sea. There should be a question to address, a method to pursue, an idea/hypothesis to test in order to write a scientific paper. None of these things are attempted in this work. Therefore, I am unfortunately recommending the rejection of the paper. However, since it seems that the main author is a PhD student, I am giving here recommendations on the actual manuscript to improve the work and perhaps focus the analysis to answer a research question. If your intention is to attract the attention of the community with an interesting dataset, then a research paper is not the right choice but instead you can use a journal that receives datasets contributions (I think wind energy science has now an option to write a dataset-type of manuscript). However, even if you decide to go that way, you need to show that your dataset is indeed attractive for this and that reason, and not simply by plotting the measurements you have acquired during your project (also you probably need to archive the dataset for community use).

**Reply:** Thank you for your valuable insights on how to improve this manuscript. The proposed changes have had a significant positive impact on the overall quality of the manuscript and how to better convey the research done. The authors really appreciate the time and attention you have put into our manuscript.

**Comment:** Introduction: you provide a very long context on the importance of offshore wind energy. Of course, the introduction must have this context, but a paragraph is more than sufficient. Also, you are providing too broad contexts and backgrounds to the subjects (for example when you talked about lidar). Even the introduction needs to be concise, precise and have only the elements a reader needs to understand your work.

**Reply:** We have modified the manuscript organisation and information, improving the readability and conciseness.

Changes: See reviewed Introduction section.

**Comment:** Figures should really be explanatory or provide results. Figure 1, as an example, does not add to the manuscript. Figure 3 could be combined with Fig. 2 (although I think you do not actually use a nacelle lidar configured as in Fig. 2b but only acquiring radial velocities at one range, so probably Fig. 2b needs to be corrected).

**Reply:** With the re-organisation of the manuscript, the lidar configuration and the location of the wind turbine schematic (originally in Figure 5) are shown in the dataset section in Figure 2. Furthermore, the lidar configuration was corrected to depict the data used in this work, removing unnecessary information.

Changes: See Figure 2.

**Comment:** In large parts of the other sections (after introduction), you again introduce and provide long context to subjects. An example (not the only one) is the first three paragraphs of Sect. 2, which are more appropriate for an introduction (In a much shorter fashion).

**Reply:** We have realised during the review process, namely with this comment, the need to improve the writing logic and overall conciseness. The Introduction has been modified and redundant information has been removed of the manuscript.

Changes: See reviewed introduction section.

**Comment:** About the power law: I really hope you are wrong and the power law is not the current industry standard for wind profiles. The power law can only be used in a climatological fashion or to extrapolate winds very locally (small vertical differences). I do not think it makes sense to give the power law the importance that you do as it is well-known by the community of the limitations of its usage.

**Reply:** The design requirements in the IEC 61400-1 and 61400-3-1 standards consider the power law with a shear exponent of 0.14 as the conditions for which to design a wind turbine. As the main goal of the study is to verify the validity of these requirements in the field, we detect moments in which this power law is not valid. The actual measured wind profiles are measured using a nacelle-mounted lidar, which are used to validate this design assumption. We clarified this by stating in the introduction that we would like to validate the design assumptions using field data. Moreover, we emphasized the limitations of the use of the power law in Section 3.1.1, with a reference to the work of Cheynet et al. (2024).

**Changes:** It reads in introduction: "Moreover, dynamic events, such as the veer and shear profiles during extreme events (e.g., storms), are automatically detected in the data, and compared to the power-law design assumptions made in the IEC standard.".

**Comment:** Repetitions: some places of your manuscript you repeat things, and it gives the impression that the version of the manuscript is not well reviewed by the co-authors. An example of this is lines 111 and 112 where you write the same twice. [ Also lines 325 vs 300]

**Reply:** We have eliminated instances in the manuscript that were inadvertently repeated.

Changes: Sentences removed.

**Comment:** Section 2.2.2 this is over descriptive and superfluous. You can just introduce the metric *(Eq. 4)* you use to evaluate veer and that is all

**Reply:** We shortened Section 3.1.2 (originally 2.2.2) and focused on the definition of veer to be clear and concise.

Changes: See Section 3.1.2.

**Comment:** Section 4.4 about the shear. If you estimated shear then you need to tell information about the vertical levels you actually use to derive it and methods. For example, if you use the power law and all the vertical levels you seem to be analyzing (20–180 m), then the computation

is wrong as the shear exponent varies with height. Please do not say things like "The wind profile is usually extrapolated using the power law"

**Reply:** We clarified in the manuscript in Section 3.1.1 that a least-squares method is used to estimate a single value for the shear over all heights, and that this is an assumption made in the analysis. We also emphasize that in the IEC 61400-3-1 design standard, a constant value of 0.14 is assumed.

**Changes:** "The measurements available for this work allows the calculation of the shear exponent between several heights along the rotor area. The shear exponent is obtained with the least square optimization methodology."

**Comment:** *Line 21 replace "gaining popularity" by "attractive"*

**Reply:** We have replaced the expression with the suggestion made in the comment.

Changes: Suggested expression "attractive" used.

**Comment:** Line 30 and similar: LiDAR is for many years a word in the AMS glossary, just like radar and so you do not need to explain the acronym and you do not need to capitalize, so it is just lidar.

**Reply:** Confronted with the information provided by the reviewer, we have decided to change the instances of "LiDAR" to "lidar", without defining the acronym.

Changes: The changes are made throughout the manuscript.

**Comment:** *Line 33 "This allows ... to be time averaged": No, the fact that you can velocity projections with a lidar does not allow for velocities to be time-averaged*

**Reply:** We have corrected the phrasing of the sentence, as it was intended to convey that the raw measurements are time- and space-averaged.

**Changes:** "Wind characteristics are then time-averaged at various spatial locations upstream and along the rotor are (...)"

**Comment:** Your references to the IEC standard, such as in line 40 need to be changed. I mean the style you use to reference those (it should not be "Commission and other, 2019" or similar)

**Reply:** We have corrected the citations that refer to the IEC standard.

Changes: In text citation now reads (IEC, 2005) and (IEC, 2019).

**Comment:** *I* guess you need to add around line 56 that you also have a section on Conclusions?

**Reply:** We have removed the last sentence, as it became clear that it is not necessary, nor a usual practice.

Changes: Sentence removed.

Comment: Line 90 the cyclops dilemma does not create ambiguities in the los measurements

**Reply:** We have corrected the phrasing as it was meant to convey that a single LOS will have an ambiguity problem called "Cyclops dilemma" and not the way around.

**Changes:** "This limitation, often referred to as the "Cyclops Dilemma," is the ambiguity that a single-LOS measurement has, making WFR challenging when relying solely on a monostatic LiDAR system (Guillemin et al., 2016)."

**Comment:** Line 94 you do not need at least 2 los, you need 3; you only need 2 if you assume something about the third los (for example that w=0)

**Reply:** After reviewing this section this sentence has been removed, as it became redundant information.

Changes: Sentence removed.

**Comment:** Some of your style to refer to work is wrong. For example, line 99 should read "... such as those in Schlipf et al. (2012) and Wagner et al., (2014),..."

**Reply:** We have corrected the style of references when needed, as the example given in the comment.

Changes: Sentences corrected.

**Comment:** *Captions on figures are weird. For example in Fig. 2 should be "Figure 2. (a) Nacelle lidar schematics or a 2-beam reconstruction method implementation. (b) A representation..."*

**Reply:** We have decided to change from (a) and (b) subfigure type by a "left and right panel" description. As an example, the "**Changes**" contains Fig. 2 new caption. Furthermore, captions have been reviewed and some rewritten to become clearer with a better description of the figure seen.

**Changes:** "Figure 2. Schematic of the wind farm where the wind turbine used during the measurement campaign is highlighted in orange with the dominant wind direction (southwest) displayed by the blue arrow in the left panel. A representation of the 64-averaged azimuthal averaged circular measurements extracted from the raw data taken at the distance used is shown in the right panel."

Comment: Similar happens in Fig. 6, for example.

**Reply:** Figure 5 (originally Figure 6) was also corrected, taking into account the new adopted format for subfigures description.

**Changes:** "Figure 5. In the left panel, the histogram with the respective Weibull distribution with the shape (k) and scale ( $\lambda$ ). The right panel shows the wind rose for the measurement campaign duration."

**Comment:** *Line 105 units and quantities should be separated by a small space*

**Reply:** We have reviewed the manuscript assuring that all units and quantities are separated by a small space; however, this specific instance has been removed after the review process.

Changes: Sentence removed.

**Comment:** Lines around line 110: how long time does it take this ZX lidar to scan the full circle? 1 s? Please rephrase so that the reader understand that the instrument provides radial velocities at 64 positions around the scanning circle.

**Reply:** After re-organising the manuscript, we have realised that this explanation would fit better under the dataset section. Furthermore, we made more explicit that the 64-averaged azimuthal position comes from a user-defined averaging and not form the device.

**Changes:** "The lidar used for the present work measures circular profiles upstream of the wind turbine at a 50 Hz frequency, with a 1-second per revolution in each plane. At each LOS measurement, the azimuthal position is registered concerning the centre of the probed circle. The raw measurements are averaged in 10-minute intervals and in space, creating a 64-averaged azimuthal position lidar, as seen in the right panel of Fig. 2.".

**Comment:** Lines around line 120: The 2-beam method assumes w=0 to infer the two horizontal velocity components. Nothing more. The lines between 120 and 133 are also very superfluous.

**Reply:** We have made the section more concise and clearer.

Changes: See reviewed manuscript.

**Comment:** *Lines* 137-138 *where did these lines come from?*

**Reply:** This sentence was inadvertently misplaced during the writing of the manuscript; so, it has been removed.

Changes: Sentence removed.

Comment: Line 147 please no this 1/7value

**Reply:** We have added a reference to this statement, making it clearer that the definition of the 1/7 power-law profile originates from the IEC standard.

**Changes:** "Furthermore, the IEC 61400-3-1 (IEC, 2019) defines the shear exponent equal to 0.14, commonly referred to as 1/7 power law profile."

**Comment:** 7 you use different sign for u and V

Reply: We have reviewed the manuscript assuring consistency in the used variables.

**Changes:** Equation 9 (originally equation 7) has the same variable in the equation and in the description.

**Comment:** Figure 4: where is this data from? You have not introduced the dataset yet

**Reply:** We have changed the dataset section to appear before the methodology chapter, to improve the reading logic of the manuscript.

Changes: Dataset section appears as Section 2 and Methodology as Section 3.

**Comment:** *Lines 210 and 211 now you change units to be in italics... they should be always normal text*

**Reply:** This instance was mistakenly inside a Latex's equation environment, we have corrected this and so, units no longer appear in italic.

Changes: Corrections made.

Comment: Line 224 a lidar or anything measures the data

Reply: After revision, this image has been removed; so, this sentence has been removed.

Changes: Sentence removed.

Comment: In Sect. 31. You should introduce the lidar, type, specification

**Reply:** We have added the missing information in the paragraph that introduces the sensor.

**Changes:** "The LiDAR deployed is a continuous-wave ZX lidar (Zephir Ltd.), installed in the nacelle of a first-row wind turbine in a wind farm in the Belgian offshore zone. (...) The lidar used for the present work measures circular profiles upstream of the wind turbine at a 50~Hz frequency, with a 1-second per revolution in each plane."

**Comment:** Caption of Fig. 5 I guess it is "wind farm" and not "wind turbine"

Reply: We have rewritten the caption for Figure 2 (originally Figure 5).

Changes: "Figure 2. Schematic of the wind farm (...)".

Comment: First 5-6 lines in Sect. 4.1 again superfluous.

Reply: We agree that this paragraph has no need for this extra information.

Changes: Sentences removed.

**Comment:** Line 174 if you provide a wind speed value then you need to say at what height it is

Reply: We have added the missing information to the sentence.

**Changes:** "For the duration of the measurement campaign, the hub-height estimated average wind speed is about 12 m/s, and the dominant wind direction is southwest (225°)."

**Comment:** Section 4.2 if you use SCADA measurements, then you need to say from which sensor specifically. If there is an estimation you do not know the origin (e.g. the so-called fit-derived method) then you cannot use it for your analysis as only god knows how this is derived.

**Reply:** We have added an extra subsection (2.2) in order to correct the oversight regarding presenting the SCADA data used in this work.

Changes: See section 2.2.

Comment: Line 308 it is Obukhov length, not Monin-Obukhov length

**Reply:** We have corrected the mistake pointed in this comment.

Changes: Corrections made.

**Anonymous Referee #3**

https://doi.org/10.5194/wes-2025-9-RC3

**Comment:** In this paper, the authors have collected LiDAR-measurements from an offshore wind turbine in Belgian waters. A method called wind field reconstruction is applied to obtain useful data from the measurements, and wind speed, wind shear profile and wind veer gradients are extracted. Additionally, SCADA-data and measurements from piles and buoys (Meetnet Vlaamse Banken) are used to validate and supplement the LiDAR results, in addition to LiDAR results directly from the manufacturer's post-processing system. The data are collected over at total of 12 months, with an interruption in the summer months. The study further investigates wind directions, wind shear profile, wind veer gradient, atmospheric stability. High shear and veer events, and atmospheric stability, turbulence intensity (from SCADA-measurements), wind-wave misalignments and differences between air temperature and sea temperature. Several events are seen where the power law profile does not fit well with the measured wind profile, for instance during storms. Moreover, veer gradients above 0.2 deg/m frequently occurs.

**Reply:** Thank you for the very insightful summary of the work done.

**Comment:** Collecting and post-processing such data is important for assessing power production and loads in wind turbines, and this paper provides valuable understandings in the applicability of LiDAR-measurements, seasonal variations in environmental conditions, and discrepancies between measured wind profiles and standardized methods. However, the paper needs considerable revision to improve readability and quality, and some issues must be clarified.

Generally, a better categorization of content in terms of introduction, methodology and results should be implemented. As of now, much of the methodology section should be moved to introduction because it describes motivation and state-of-the-art rather than the methodology. The author is invited to think through the purpose of each section before revising.

**Reply:** Thank you for your comments on how to improve the manuscript, as they are of great importance to improve the transmission of information about the work done. Furthermore, your specific comments on the manuscript have been very helpful for this revision process, and the authors are very thankful for your effort revising this manuscript.

**Comment:** *Title: The title is misleading, as the actual impacts of the wind profiles on turbine performance and response is not assessed in this work.*

**Reply:** After the revision, and as this comment points out, we believe that indeed the title unsatisfactorily represents the work done.

**Changes:** New title: Assessing wind profiles in an offshore wind farm site for field data-enabled design

**Comment:** Abstract: In general, the abstract should be more specific about the methods and results of the paper. It now consists of many diffuse sentences such as "The findings align with the current

literature on the correlation between events and weather conditions and the clear difference between wind profiling and a power law wind profile for loads design as proposed by IEC".

**Reply:** After conducting the review considering all comments made by the referees, we have, in part, rewritten the abstract.

Changes: Please see reviewed abstract.

**Comment:** Line 43: "Such data reflects the actual atmospheric conditions, including localised effects of atmospheric stability, turbulence, and wind direction variability, which are essential for accurately assessing turbine performance and loading" Can you add some references discussing turbine performance and loading under these effects?

**Reply:** We have re-structured the Introduction, so this paragraph has slightly changed. However, this information was more or less kept in a different paragraph, and we also have added.

**Changes:** "Wind profiles, which are defined by parameters such as shear, veer, and turbulence intensity, directly impact energy production estimates (Wagner et al., 2009; Murphy et al., 2019), turbine loading (Dimitrov et al., 2015; Gualtieri, 2016; Robertson et al., 2019), and ultimately, structural longevity."

Comment: Line 44-45: Are real-life conditions not usual?

**Reply:** The sentence regarding about "*real-life conditions [are] not usual*" was removed, as it did not correctly convey what we want to transmit in this section.

Changes: Information removed from paragraph.

**Comment:** *Line 47: "and the factors for turbine load design are analysed". What does this mean? There is no evaluation on influence on turbine loads in this work.*

**Reply:** We agree that how it was originally written, it could lead to misunderstandings on the work done.

**Changes:** "The proposed framework extracts wind parameters, including shear, veer, turbulence intensity, and atmospheric stability. Thus, it becomes possible to refine the wind profile, ensuring that wind turbines might be further optimised for the conditions at a specific site (Ziegler et al., 2015; Dimitrov et al., 2019)."

**Comment:** *Line 49: "validate key insights on the long-term steady state environmental conditions..."* Do we have long-term steady state conditions in the wind?

**Reply:** As this phrasing poorly reflects what we wanted to say in this paragraph, we have removed it entirely. To add that the Introduction section was largely rewritten during the review process. We have added to this reply what substituted the commented line.

**Changes:** "This paper proposes a methodology for site-specific wind profile characterisation using nacelle-mounted lidar, SCADA data, and environmental measurements from the Meetnet Vlaamse Banken (MVB) monitoring network."

**Comment: Line 54: Be consistent with "Sect". vs "Section"**

**Reply:** Due to the manuscript rules, as follows: "The abbreviation "Sect." should be used when it appears in running text and should be followed by a number unless it comes at the beginning of a sentence.". However, this comment highlighted that the phrasing can be improved.

**Changes:** "The remainder of this paper is structured as follows: Sect. 2 introduces the datasets used; Sect. 3 details the wind profile estimation, environmental classification, and profile detection methodology; Sect. 4 presents the main findings and discusses their relevance for load assessment and site-specific optimisation."

**Comment:** Figure 1: It is not clear what this figure is intended to represent. Does it include data measurements/field data? The text with the reference (line 43-45) and the caption gives opposite impressions. And what is meant by field-enabled design validation?

**Reply:** The figure intends to show how field-data from a "Production fleet" can be introduced in the "traditional" design analysis. We mean that the design can be further validated using real-life wind profiles that can be obtained using field data. For this purpose, we have corrected the caption as well as improved the explanation in the text.

**Changes:** "Figure 1. Diagram showing the traditional design analysis (blue dashed rectangle) where the design load cases are used by the simulation models, which in turn are validated with field (top right) and experimental (top left) tests. With the production fleet information, i.e. field data from in-situ measurements, creating a field-enabled design validation (green dashed rectangle)."

**Comment:** *Lines 58 to 61: Would be better off in the introduction. Same goes for lines 70-83. Wind veer should be defined early, as it is not as known as wind shear and turbulence intensity (not a typical design requirement).*

**Reply:** The manuscript was thoroughly reviewed and information present in the sections of dataset and methodology was moved to the introduction, as this comment suggests.

Changes: See reviewed Introduction section.

**Comment:** *Line* 87: *the abbreviation WFR has already been stated.*

**Reply:** We have corrected the repetition of the abbreviation's definition.

Changes: "WFR methods are techniques designed to (...)"

**Comment:** Figure 2: The LOS-abbreviations are not defined in the text.

**Reply:** After the re-organisation of the manuscript, the definition of this abbreviation appears before the caption of Figure 3 (originally Figure 2).

Changes: Abbreviation is now defined before being used in the mentioned Figure.

**Comment:** *Line 112: "in the first row of the wind farm" is repeated.*

**Reply:** We have removed the repeated part of the sentence.

**Changes:** "Furthermore, the chosen wind turbine is located in the first row of the wind farm, such that for a set of wind directions, the incoming wind field is undisturbed."

**Comment:** *Chapter 2.2: It is unfortunate that alpha is used both to describe the LiDAR setup and the shear exponent*

**Reply:** We made sure that all variables are now distinguished by different Greek letters.

**Changes:**  $\gamma$  is used for the lidar's opening angle,  $\Delta$  for the vectors' misalignment,  $\theta$  for wind direction and  $\alpha$  for shear exponent.

**Comment:** *Line 120: It is not clear how Vx and Vy are obtained.*

**Reply:** We have added the formulas used for the calculation of mentioned variables.

Changes: Eqs (1) and (2) were added.

**Comment:** *Lines* 138-139: *It seems this sentence is misplaced as it has no direct connection to the previous text.*

**Reply:** We have removed this sentence, that was in fact misplaced and had no connection with this section of the manuscript. We would like to highlight our appreciation for your effort in reviewing our manuscript.

Changes: Sentence removed.

**Comment:** Lines 143-144: "it has been proven that it cannot be as correct since shear is affected by numerous factors, such as turbulence intensity, atmospheric stability and surface roughness." I think it would be more meaningful to say that TI and shear are measures of the atmospheric stability rather than that shear depend on TI. I would also recommend looking at the work by Olsen et al.: "Evaluation of Marine Wind Profiles in the North Sea and Norwegian Sea Based on Measurements and Satellite-Derived Wind Products" (2022).

**Reply:** The phrasing was not correctly made as we did not want to convey that the shear depends on TI. We have corrected this paragraph to transmit what we wanted, which was the fact that the wind profile differs in certain occasions with the power law profile that is proposed in the standard. Furthermore, thank you for the very relevant reference for our work.

**Changes:** "Although power law is normally used to extrapolate wind speed, it has been proved that it can differ from site- and event-specific profiles (Cheynet et al., 2024)."

**Comment:** *Line 147: At first, it was a bit unclear that it is the power law exponent that is set to* 0.14 *in most (but not all?) of the attempts to fit the LiDAR wind profiles to the power law.*

**Reply:** We have added to the subsection further context and explanation which makes it clearer that all profiles estimated are indeed compared with a power law profile with a constant shear of 0.14, as it is proposed in the IEC standard. Furthermore, this information was also added in the

introduction section, to make it clearer from the beginning of the manuscript. The changes added here correspond to the sentence added in the introduction.

**Changes:** "Furthermore, dynamic wind events are automatically detected by fitting measured profiles against the standard IEC 61400-1 power-law profile, which proposed the use of a constant 0.14 shear exponent (IEC, 2005)."

**Comment:** *Line 152: Gao refers directly to another source when stating this – it would be better to refer to the initial/base source.*

**Reply:** We have added the original work performed by Holton and Hakim, 2013, which is, as you correctly mentioned, the base source for Gao et al. (2019).

Changes: Citation added: "Holton, J. R. and Hakim, G. J.: Chapter 8 - The Planetary Boundary Layer, pp. 255–277, Academic Press, Boston, fifth edn., https://doi.org/10.1016/B978-0-12-384866-6.00008-8, 2013."

**Comment:** *Line 154: Shu et al does not investigate loads and performance – why are they referred here?*

**Reply:** Shu et al. (2020) was incorrectly cited, we have reviewed the references and rewritten part of the paragraph to be clearer.

**Changes:** "Wind veer will influence the wind field that reaches the wind turbine, impacting wake deflection (Churchfield and Sirnivas, 2018; Brugger et al., 2019) and power production (Bardal et al., 2015; Gao et al., 2021)"

**Comment:** *Equation (4): The definition of the bulk veer gives an absolute value – how do you differ between veer and backing?*

**Reply:** Equation 6 (originally equation 4) was mistakenly added with the module symbols, whereas is calculated without, in order to obtain negative and positive values.

Changes: See Eq. 6.

Comment: Line 160-164 is a replica of line 155-157.

**Reply:** We have removed the mistakenly repeated sentence.

Changes: Removed first appearance instance.

Comment: Line 175-175: sigma u or sigma V?

**Reply:** It should read sigma\_V, as in the equation 9 (originally equation 7); the typo is corrected.

**Changes:** "where  $\sigma_V$  is the standard deviation and  $\overline{V}$  the mean of the wind speed".

**Comment:** *Line 181-182: This sentence is not complete and should be reviewed. ("being only in near neutral conditions that this profile usually occurs")*

**Reply:** We have corrected the incorrect phrasing of the sentence. Furthermore, due to the review made for the atmospheric stability results, the majority of paragraph has been rewritten. The changes added here are with respect to the information regarding the power law profile and atmospheric stability relation.

**Changes:** "Furthermore, the power law profile mostly occurs under near neutral conditions (Sakagami et al., 2015), so non-fitting profiles are expected to be found in atmospheres either with higher (unstable conditions) or lower (stable conditions) mixing conditions."

Comment: Line 184: "Length" is misspelled.

**Comment:** *Line 185: There is a dot between the sentence and the reference (Holtslag).*

**Reply:** We have corrected the typos mentioned in your review.

Changes: Typos corrected.

**Comment:** Eq (8): A temperature parameter  $(273.15 + T_a)$  is missing in the denominator (ref Albornoz et al. who is referring to Bahamonde and Litran). Please clarify whether this is a typo, or if this is the equation applied in the analyses.

**Reply:** The original equation has the temperature given in degree Celsius, but for the current work, the temperature is given in Kelvin, so the term 273.15 is not necessary. We improved the explanation that accompanies equation 10 (originally equation 8) and we cite the original work from Ebuchi et al. (1992).

**Changes:** "(...) it is possible to calculate the Bulk Richardson Number, as in (Ebuchi et al., 1992), with:" and "(...) the air and sea temperatures in Kelvin and (...)"

**Comment:** *Eq* (8): *Why not use the same symbols for*  $T_air$  *and*  $T_sea$  *as in equation 6? And where is z taken? At the hub-height?*

**Reply:** We have corrected the manuscript so that variables that represent the same quantities, such as in equations 8 and 10 (originally equations 6 and 8), are the same. The z is the height where the measurements are made, information conveyed in the dataset section, which is now before the methodology.

Changes: Information moved to be before presenting equation 10 (originally equation 8).

**Comment:** Equation (9) and this section: How is the Obhukov length used for categorization? That is, what are the criteria for being categorized as "stable", "neutral", etc?

**Reply:** Due to a lapse, the categories were only defined in the results chapter. After reviewing the manuscript, we added this information in the methodology as Table 1.

**Changes:** Table 1 has been added and the following "After obtaining L, it is possible to define seven categories depending on the interval shown in Table 1, based on the works of (Sathe et al., 2010; Holtslag et al., 2014):"

Comment: Line 189: "the air and ... temperatures" "sea" is missing.

**Reply:** We have added the missing word.

**Changes:** "(...) the air and sea temperatures in Kelvin (...)".

**Comment:** *Line 200: It would be more clear if it was explicitly stated that alpha is taken as 0.14 (if that is the case).*

**Reply:** We have improved the paragraph by adding more and clearer context.

**Changes:** "Power-law wind profiles fail to capture some events, such as high wind speeds and rapid wind direction change (Debnath et al., 2021). For the current paper, events will be detected based on the fitting of the estimated profile with the 1/7 power-law profile, which has a constant shear exponent of 0.14. For the current work, these are considered generic events with wind profiles that fail to fit the power-law, instead of being detected according to their meteorological definitions, e.g. low-level jets in Shu et al. (2018). The implemented methodology enables the detection of more events, since not all deviations from the power-law adhere to a specific meteorological definition. The detection is made using the mean absolute error (MAE) between the estimated and 1/7 power-law profiles, defined as:

**Comment:** *Line 203-204: What do you mean by "non-correlated" here, and what is a "weather event"?*

**Reply:** It was meant to explain that we will study the detected profiles that differ from the powerlaw profile but the "non-correlated" expression is not correctly used; the "weather" has been removed as it does not add relevant information. The sentence has also been moved to the end of the subsection

**Changes:** "Furthermore, the non-power-law profiles will be related to registered storms (e.g. Ciarán in November of 2023) in the studied location."

**Comment:** *Lines 215-220: It would have been useful to read something like this overview of the dataset earlier, perhaps in the introduction (in the paragraph starting with "In this study, a met..."*

**Reply:** We have moved the dataset to be before the methodology section, which improves the reading logic of the manuscript.

Changes: See reviewed manuscript.

**Comment:** *Line 206: Is the extreme direction change a spatial or temporal change?*

**Reply:** The EDC defined by the IEC is the change in wind direction  $(\theta_0 + \theta_e)$  that has a duration of 6 seconds. We have rewritten the paragraph to be clearer; namely better introducing the definition in the beginning of the subsection.

**Changes:** "The IEC 61400-1 (IEC, 2005) standard defines the extreme direction change (EDC) events by the initial wind direction,  $\theta_0$ , change in degrees for a specific duration, T, equal to 6 seconds."

Comment: Line 210-211: Why are units in italics?

**Reply:** The units were added inside the equation environment and stayed in italic. We have reviewed the manuscript to assure it does not happen.

Changes: Units are not in italic.

**Comment:** In general, it would be better to place this section [dataset] before the Methodology-section, or to implement it in the beginning of the Methodology-section.

**Reply:** We have moved the dataset section to be before the methodology, as we understand that highly improves the reading logic.

Changes: See reviewed manuscript.

**Comment:** *Line 216: It is stated that three data sources are discussed, but only the LiDAR and Meetnet Vlaamse Banken data are discussed in the following sections.*

**Reply:** There was an oversight in introducing the SCADA data, so we have added the correspondent explanation to the Dataset section.

Changes: See subsection 2.2.

**Comment:** *Line 219: Do you have a reference to the Meetnet Vlaamse Banken data?*

**Reply:** We have added the reference in the beginning of the section.

Changes: "(...) the Meetnet Vlaamse Banken data (en Kust, 2024)."

Comment: Line 226: What is meant by "heights for hub height"?

**Reply:** We have corrected the strange phrasing of the sentence as it was intended to convey that the data is available for several heights along the rotor area.

**Changes:** "The deployed lidar provides a model based fit-derived (FD) wind components for the 10-minute averaged dataset at several distances in front on the sensor and heights along the rotor area."

**Comment:** *Line 240: Refers to Section 3, but this is also placed in Section 3.*

**Reply:** During the review process, this sentence has been removed, but the comment highlighted the relevance of reviewing these references in the manuscript

Changes: Sentence removed during the review process.

**Comment:** *Line 244: What does this mean: "However, the absence of wake data presents a limitation for wind ..."*?

**Reply:** We have phrased better that one of the limitations of the implemented methodology is the fact that it only analyses within the freestream region, due to the nature of the assumption of the used reconstruction method.

**Changes:** "This directional filtering is necessary to isolate relevant wind conditions, as non-freestream measurements would negatively influence the accuracy of the reconstruction method (Marini et al., 2024). This limits the current investigation to the non-wake, or freestream, region."

**Comment:** *Line 256: "These results are compared with the distribution of events" – what is meant by "distribution of events"?*

**Reply:** It was meant to create a bridge with the data presented and how it's going to be used; after reviewing it, we have realised it is not very clear.

Changes: Sentence removed.

**Comment:** *Line 262: "time series compared with SCADA and the manufacturer's reconstruction." What manufacturer?*

**Reply:** We have added further information to be clear that it is the lidar's manufacturer that have a proprietary (i.e. black-box) reconstruction.

**Changes:** "This is followed by validating the reconstruction method with a time series comparison with SCADA and the LiDAR manufacturer's proprietary wind speed reconstruction."

**Comment:** Figure 7: The caption refers to a "first row (a) and a second row (b)". Is this correct?

**Reply:** We have corrected the caption of Figure 6 (originally Figure 7), as it was erroneously present in the manuscript.

**Changes:** "Seasonal wind roses of the measured wind data for non-consecutive months between February 2023 and March 2024."

Comment: Line 294: Be consistent with using "2-beam" or "2-Beam".

**Reply:** We have assured the consistency with "2-beam" throughout the manuscript.

Changes: The use of "2-beam" confirmed in the manuscript.

**Comment:** *Line* 306-309: *This should be in the introduction.*

**Reply:** Repeated information removed since it is presented in the Methodology section.

Changes: Information removed from results section.

**Comment:** Figure 9: For this figure to be useful in terms of seasonal variation, it should show the name of the months.

**Reply:** We have added the months to all figures that this comment applies.

Changes: See Figures 9, 11 and 15.

**Comment:** Line 310: "...the figure indicate that neutral conditions dominate...". To me, based on Figure 9, it looks like very unstable conditions dominate in all months except month 8 and 10. Please clarify.

**Reply:** In the explanation, it was missing to include the quasi-neutral atmospheric conditions. Taking into account the neutral-stable and neutral-unstable conditions, the statement becomes clearer and more correct.

**Changes:** "The results in the figure indicate that quasi- and neutral conditions dominate the atmospheric stability distribution over the course of the campaign."

**Comment:** Lines 315-320: According to the authors, stable and very stable conditions are seen more frequently in the colder months. This explained by colder air moving over warmer water. However, winter, and this temperature difference is typically associated with unstable conditions, and summer, with warmer air over colder water is associated with stable conditions (see. e.g. Maarten Holtslag's phd thesis "Far offshore wind conditions in scope of wind energy"). Please clarify. You discuss this in lines 351-355 too, with the opposite conclusion than in lines 315-320. I believe the discussion in lines 351-355 is more correct.

**Reply:** In other literature, such as in "Sathe, A., Gryning, S. E., & Peña, A. (2011). Comparison of the atmospheric stability and wind profiles at two wind farm sites over a long marine fetch in the North Sea. Wind Energy, 14(6), 767-780" the description of the results indicates that during summer months there is more unstable atmospheric conditions, while in winter months there is more stable. The discussion has now been altered to reflect only on the obtained results, with some corroboration from other literature, but avoiding commenting on a general conclusion about this topic.

Changes: See reviewed discussion for shear (lines 298-307) and veer (lines 323-327).

**Comment:** Lines 321-323: For the offshore location OWEZ, Sathe found that neutral conditions dominated in high wind speeds, but this was not the case below say 10-11 m/s. Similar trends were also found for the other three locations. I think this generalization of their results is erroneous. Do you have other resources supporting the findings that neutral conditions dominate offshore? Referring also to the statement in lines 311-312.

**Reply:** We apologise as the incorrect reference has been cited for this analysis. The correct reference should be "Sathe, A., Gryning, S. E., & Peña, A. (2011). Comparison of the atmospheric stability and wind profiles at two wind farm sites over a long marine fetch in the North Sea. Wind Energy, 14(6), 767-780", where the findings of a location near the studied one resulted in the same conclusions. However, the hard statement about being "dominant offshore" has been removed, as it poorly transmits the authors' intention, as the discussion should focus on the obtained results and not in a generalization for offshore conditions and locations.

Changes: See reviewed discussion between lines 265 and 280.

**Comment:** Line 332-333: "...if the power law is assumed, the IEC standard will be more conservative in the majority of cases." Is it conservative to assume high shear? Conservative for the power production? For the turbine loads?

**Reply:** Yes, as the assumption of higher wind speeds at hub height can lead to load and power prediction estimations larger than the reality. Information has been added to the sentence to increase clarity.

**Changes:** "Around 87% of the estimated shear will fall between zero and the 0.14 value, meaning if the 1/7 power-law profile is used, a higher wind speed gradient is unnecessarily assumed."

**Comment:** *Line 335: this phrase "which implies a non-power law profile the large absolute value it is" should be rephrased.*

**Reply:** We have rephrased to make the sentence clearer by removing this piece of information, which is, at this point of the manuscript, slightly out-of-place.

Changes: Sentence removed.

**Comment:** Line 339-340: "... it is possible to reconstruct the wind speed at several heights, so it is also possible to calculate the value of the power exponent using E1q. 3, in which the assumption of the power law profile is taken." Wasn't this also done when calculating the average shear coefficient in line 330? Why is it mentioned here?

**Reply:** The authors felt it was relevant to highlight this information but we have realised it might be at this point of the manuscript a redundancy that decreases readability. So, the sentence has been removed.

Changes: Sentence removed.

**Comment:** Figure 11: Again, it would be useful to label the months (or at least categorize them into "summer", "spring", "winter"...

Reply: Months added to all figures that this comment applies.

Changes: See Figures 9, 11 and 15.

**Comment:** *Line* 346-347: *This sentence is not complete ("While wind-wave..")*

**Reply:** We have corrected the grammar of the sentence.

**Changes:** "Wind-wave misalignment, or the difference (...)"

**Comment:** *Line 361: Can veer be both positive and negative? Or does backing have negative and veer positive gradients?*

**Reply:** Yes, there are authors that refine the definition on veer with the hub height (https://doi.org/10.1063/5.0033826). The authors felt no necessity of that analysis for the current work - so the veer is calculated as a gradient from the lowest height to the highest height.

Changes: No action made.

**Comment:** *Line* 371: *What is the unit of the veer value of* 0.2? *What categories do you refer to?*

**Reply:** We have added the veer units, which is degree per meter, and the categories of the plot have been clarified.

**Changes:** "In Fig. 15, the monthly occurrences of high veer, i.e., absolute value above 0.2 [°/m], and separated by the categories of above-rated power and between rated and cut-in wind speeds, are shown."

**Comment:** Line 371-375: Be careful with assuming a causal relationship between veer and these "weather parameters". While there can be a correlation between turbulence intensity and high veer, it doesn't mean that turbulence intensity is influencing veer.

**Comment:** Lines 374-375: I am not sure if you are discussing the results in Figure 16c (e.g. the correlation between high veer and turbulence intensity) or discussing the uncertainties in the measurements.

**Reply:** We have improved the phrasing to be clearer that there is no attempt in creating a relationship between parameters, but yes to demonstrate that there is a commonality between them.

Changes: See lines 318 to 326.

**Comment:** *Lines 380-385: I would recommend referring to Julie Lundquists chapter 27 "Wind shear and wind effects on wind turbines" in "Handbook of Wind Energy Aerodynamics".*

**Reply:** We thank the reviewer for the suggested reference. However, as we want to keep the focus on discussing performed relevant studies, we decided to remove the part discussing in-depth theory about veer.

Changes: See lines 327 to 331.

**Comment:** *Lines* 385-390: *I think this would fit better in the introductory chapter, at is considers motivation for assessing veer.*

**Reply:** We have removed from here as it was slightly repeated information, and kept in the Introduction as the comment suggests.

Changes: Information removed from Results section.

**Comment:** *Lines* 398-400: *You show an example of a storm – should this not be put under section* "4.6.1 *Storms*"?

**Reply:** We agree that a separate subsection on storms may cause confusion, as both sections 4.6 and 4.6.1 focus on detecting non-power-law events while using storms dates for the hypothesis that these events can have higher probability of non-power-law profiles. Therefore, we decided to combine the results in a single section called 'Non-power-law events'.

Changes: Subsection 4.6.1 joined with 4.6.

**Comment:** *Line 405: "it is evident that the IEC standard assumption holds for most of the day, as seen in figure 10". How does figure 10 and 19 support this?*

Reply: The figure label was wrongly chosen in Latex; it has been now corrected.

**Changes:** "(...) it is evident that the IEC standard assumption holds for most of the day, as seen in even during a storm as in Fig. 19."

**Comment:** *Line 414: "appear unrelated" could be replaced by "is not possible to describe by the power law"*

Reply: We have corrected the sentence inspired by the suggested replacement.

**Changes:** "However, during this period, there was an increase in the average wind speed, and the power law profile cannot describe some of the estimated wind profiles."

**Comment:** *Line 415: This is not a complete sentence.*

**Reply:** We have corrected the grammar of the sentence.

Changes: "Figure 21 shows a portion of the Jocelyn (in January of 2024) storm."

**Comment:** *Line 418: "has very little similarity to the power law profile" would be better.*

**Reply:** We have corrected the sentence inspired by the suggested replacement.

**Changes:** "However, the profiles during Jocelyn have less similarities with the standard power law than in Ciáran."

**Comment:** Line 426: Why is wind field reconstruction unavailable during the EDCs?

**Reply:** Due to the filtering of valid LOS measurements from raw data. We have added context to the paragraph to be clearer.

**Changes:** "Wind field reconstruction is sometimes unavailable during these short-duration EDCs, as it might not valid due to a large misalignment between incoming wind field and LOS measurement."

**Comment:** *Line 36: NREL ".n.d." – what does this mean?*

**Reply:** The citation was initially added without the year, which lead to adding "n.d." as it stands for "non-disclosed"; however, this comment highlights the necessity of correcting the citation, which now has the year that the webpage has been last updated.

**Changes:** "National Renewable Energy Laboratory: Dynamometer Research Facilities, https://www2.nrel.gov/wind/facilities-dynamometer, [Online; accessed April 08, 2025], 2025." and in text citation reads "(National Renewable Energy Laboratory, 2025))"

**Comment:** Generally, the references to the IEC-standards seem off (they refer to someone named "Comission et al."

**Reply:** We have corrected both IEC references to be more up-to-standard.

Changes: In text citation now reads (IEC, 2005) and (IEC, 2019).

**Comment:** Use \noindent to prevent a new paragraph after an equation (if the following text belongs to the same paragraph as the equation).

Reply: We would like to thank you for the tip on how to easily implement the necessary correction.

**Changes:** We have made the correction throughout the manuscript.

Comment: Avoid lonely subsections (e.g. 2.3.1, 2.4.1, 3.1.1, 4.1.1, 4.4.1).

**Reply:** As it became more evident with the review process, and as this comment sharply highlights, to improve the writing logic of the manuscript, lonely subsections were eliminated.

Changes: See reviewed manuscript.

**Comment:** For citations referring directly to the author, the author name should not be in parentheses. E.g. line 235: "As described in (Marini et al., 2024)" should be "As described by Marini et al. (2024)". In latex, using \cite{} instead of \citep{} resolves this issue.

**Reply:** Once again, we appreciate the tip to easily implement the necessary corrections.

Changes: We have made the correction throughout the manuscript.